# FOURIER REPRESENTATIONS FOR BLACK-BOX OPTIMIZATION OVER CATEGORICAL VARIABLES

## ABSTRACT

Optimization of real-world black-box functions defined over purely categorical variables is an active area of research. In particular, optimization and design of biological sequences with specific functional or structural properties have a profound impact in medicine, materials science, and biotechnology. Standalone acquisition methods, such as simulated annealing (SA) and Monte Carlo tree search (MCTS), are typically used for such optimization problems. In order to improve the performance and sample efficiency of such acquisition methods, we propose to use existing acquisition methods in conjunction with a surrogate model for the black-box evaluations over purely categorical variables. To this end, we present two different representations, a group-theoretic Fourier expansion and an abridged one-hot encoded Boolean Fourier expansion. To learn such models, characters of each representation are considered as experts and their respective coefficients are updated via an exponential weight update rule each time the black box is evaluated. Numerical experiments over synthetic benchmarks as well as real-world RNA sequence optimization and design problems demonstrate the representational power of the proposed methods, which achieve competitive or superior performance compared to state-of-the-art counterparts, while improving the computational cost and/or sample efficiency substantially.

## 1 INTRODUCTION

A plethora of practical optimization problems involve black-box functions, with no simple analytical closed forms, that can be evaluated at any arbitrary point in the domain. Optimization of such black-box functions poses a unique challenge due to restrictions on the number of possible function evaluations, as evaluating functions of real-world complex processes is often expensive and time consuming. Efficient algorithms for global optimization of expensive black-box functions take past queries into account in order to select the next query to the black-box function more intelligently. While black-box optimization of real-world functions defined over integer, continuous, and mixed variables has been studied extensively in the literature, limited work has addressed incorporation of purely categorical type input variables.

Categorical type variables are particularly challenging when compared to integer or continuous variables, as they do not have a natural ordering. However, many real-world functions are defined over categorical variables. One such problem, which is of wide interest, is the design of optimal chemical or biological (protein, RNA, and DNA) molecule sequences, which are constructed using a vocabulary of fixed size, e.g. 4 for DNA/RNA. Designing optimal molecular sequences with improved or novel structures and/or functionalities is of paramount importance in material science, drug and vaccine design, synthetic biology and many other applications (see Dixon et al. (2010); Ng et al. (2019); Hoshika et al. (2019); Yamagami et al. (2019)). Design of optimal sequences is a difficult black-box optimization problem over a combinatorially large search space (Stephens et al. (2015)), in which function evaluations often rely on either wet-lab experiments, physics-inspired simulators, or knowledge-based computational algorithms, which are slow and expensive in practice. Another problem of interest is the constrained design problem, e.g. find a sequence given a specific structure (or property), which is inverse of the well-known folding problem discussed in Dill & MacCallum (2012). This problem is complex due to the strict structural constraints imposed on the sequence. In fact one of the ways to represent such a complex structural constraint is to constrain the next choice

sequentially based on the sequence elements that have been chosen a priori. Therefore, we divide the black box optimization problem into two settings, depending on the constraint set: $(i)$ Generic Black Box Optimization (BBO) problem referring to the unconstrained case and $(ii)$ Design Problem that refers to the case with complex sequential constraints.

Let $x_t$ be the $t$-th sequence evaluated by the black box function $f$. The key question in both settings is the following: Given prior queries $x_1, x_2 \ldots x_t$ and their evaluations $f(x_1) \ldots f(x_t)$, *how to choose the next query $x_{t+1}$?* This acquisition must be devised so that over a finite budget of black-box evaluations, one is closest to the minimizer in an expected sense over the acquisition randomness.

In the literature, for design problems with sequential constraints, MCTS (Monte Carlo Tree Search) based acquisitions are often used with real function evaluations $f(x_t)$. In the generic BBO problems in the unconstrained scenario, Simulated Annealing (SA) based techniques are typically used as acquisition functions. A key *missing* ingredient in the categorical domain is a surrogate model for the black-box evaluations that can interpolate between such evaluations and use *cost-free* approximate evaluations from the surrogate model internally (in acquisition functions) in order to reduce the need for frequently accessing real evaluations. This leads to improved sample efficiency in acquisition functions. Due to the lack of efficient interpolators in the categorical domains, existing acquisition functions suffer under a finite budget constraint, due to reliance on only real black-box evaluations.

**Contributions**: We address the above problem in our work. Our main contributions are as follows:

1. We present two representations for modeling real-valued combinatorial functions over categorical variables, which we then use in order to learn a surrogate model for the generic BBO problem and the design problem. The surrogate model is updated via a hedge algorithm where the basis functions in our representations act as experts. The latter update happens once for every real black-box evaluation. To the best of our knowledge, the representations and/or their use in black-box optimization of functions over categorical variables are novel to this work.
2. In the BBO problem, the proposed method uses a version of simulated annealing that utilizes the current surrogate model for many internal cost-free evaluations before producing the next black-box query.
3. In the design problem, the proposed method uses a version of MCTS in conjunction with the current surrogate model as reward function of the terminal states during intermediate tree traversals/backups in order to improve the sample efficiency of the search algorithm.
4. Numerical results, over synthetic benchmarks as well as real-world biological (RNA) sequence optimization and design problems demonstrate the competitive or superior performance of the proposed methods over state-of-the-art counterparts, while substantially reducing the computation time and sample efficiency, respectively.

## 2 RELATED WORK

Hutter et al. (2011) suggests a surrogate model based on random forests to address optimization problems over categorical variables. The proposed SMAC algorithm uses a randomized local search under the expected improvement acquisition criterion in order to obtain candidate points for black-box evaluations. Bergstra et al. (2011) suggests a tree-structured Parzen estimator (TPE) for approximating the surrogate model, and maximizes the expected improvement criterion to find candidate points for evaluation. For optimization problems over Boolean variables, multilinear polynomials Ricardo Baptista (2018); Dadkhahi et al. (2020) and Walsh functions Leprêtre et al. (2019) have been used in the literature.

Bayesian Optimization (BO) is a commonly used approach for optimization of black-box functions (Shahriari et al. (2015)). However, limited work has addressed incorporation of categorical variables in BO. Early attempts based on converting the black-box optimization problem over categorical variables to that of continuous variables have not been very successful (Gómez-Bombarelli et al. (2018); Golovin et al. (2017); Garrido-Merchán & Hernández-Lobato (2020)). A few BO algorithms have been specifically designed for black-box functions over combinatorial domains. In particular, the BOCS algorithm Ricardo Baptista (2018), primarily devised for Boolean functions, employs a sparse monomial representation to model the interactions among different variables, and uses a sparse Bayesian linear regression method to learn the model coefficients. The COMBO algorithm of Oh et al. (2019) uses Graph Fourier Transform (GFT) over a combinatorial graph, constructed via graph cartesian product of variable subgraphs, to gauge the smoothness of the black-box function. However,

both BOCS and COMBO are hindered by associated high computational complexities, which grow polynomially with both the number of variables and the number of function evaluations.

More recently, a computationally efficient black-box optimization algorithm (COMEX) (Dadkhahi et al. (2020)) was introduced to address the computational impediments of its Bayesian counterparts. COMEX adopts a Boolean Fourier representation as its surrogate model, which is updated via an exponential weight update rule. Nevertheless, COMEX is limited to functions over the Boolean hypercube. We generalize COMEX to handle functions over categorical variables by proposing two representations for modeling functions over categorical variables: an abridged one-hot encoded Boolean Fourier representation and Fourier representation on finite Abelian groups. The utilization of the latter representation as a surrogate model in combinatorial optimization algorithms is novel to this work. Factorizations based on one-hot encoding has been previously (albeit briefly) suggested in Ricardo Baptista (2018) to enable black-box optimization algorithms designed for Boolean variables to address problems over categorical variables. Different from Ricardo Baptista (2018), we show that we can significantly reduce the number of additional variables introduced upon one-hot encoding, and that such a reduced representation is in fact complete and unique.

For design problems, we focus on the RNA sequence design problem (RNA inverse folding). The goal is to find an RNA sequence consistent with a given secondary structure, as the functional state of the RNA molecule is determined by the latter structure (Hofacker et al. (1994)). Earlier RNA design methods explore the search space by trial and error and use classic cost function minimization approaches such as adaptive random walk (Hofacker (2003)), probabilistic sampling (Zadeh et al. (2011)), and genetic algorithms (Taneda (2015)). Recent efforts employ more advanced machine learning methods such as different Monte Carlo Tree Search (MCTS) algorithms, e.g. MCTS-RNA (Yang et al. (2017)) or Nested MCTS (Portela (2018)), and reinforcement learning that either performs a local search as in Eastman et al. (2018) or learns complete candidate solutions from scratch (Runge et al. (2018)). In all these approaches, the assumption is that the algorithm has access to a large number of function evaluations, whereas we are interested in sample efficiency of each algorithm.

As an alternative to parameter free search methods (such as SA), Swersky et al. (2020) suggests to use a parameterized policy to generate candidates that maximize the acquisition function in Bayesian optimization over discrete search spaces. Our MCTS acquisition method is similar in concept to Swersky et al. (2020) in the sense that the tabular value functions are constructed and maintained over different time steps. However, we are maintaining value functions rather than a policy network.

## 3 BLACK-BOX OPTIMIZATION OVER CATEGORICAL VARIABLES

**Problem Setting**: Given the combinatorial categorical domain $\mathcal{X} = [k]^n$ and a constraint set $\mathcal{C} \subseteq \mathcal{X}$, with $n$ variables each of cardinality $k$, the objective is to find

$$x^* = \arg\min_{x \in \mathcal{C}} f(x) \tag{1}$$

where $f : \mathcal{X} \mapsto \mathbb{R}$ is a real-valued combinatorial function. We assume that $f$ is a black-box function, which is computationally expensive to evaluate. As such, we are interested in finding $x^*$ in as few evaluations as possible. We consider two variations of the problem depending on how the constraint set $\mathcal{C}$ is specified.

**Generic BBO Problem:** In this case, the constraint set $\mathcal{C} = \mathcal{X}$. For example, RNA sequence optimization problem that searches for an RNA sequence with a specific property optimized lies within this category. A score for every RNA sequence, reflecting the property we wish to optimize, is evaluated by a black box function.

**Design Problem:** The constraint set is complex and is only sequentially specified. For every sequence of $x_1 x_2 \ldots x_i$ consisting of $i$ characters from the alphabet $[k]$, the choice of the next character $x_{i+1} \in \mathcal{C}(x_1 x_2 \ldots x_i) \subseteq [k]$ is specified by a constraint set function $\mathcal{C}(x_1 \ldots x_i)$. The RNA inverse folding problem in Runge et al. (2018) falls into this category, where the constraints on the RNA sequence are determined by the sequential choices one makes during the sequence design. The goal is to find the sequence that is optimal with respect to a pre-specified structure that also obeys complex sequential constraints.

**Our Techniques**: In order to address this problem, we adopt a surrogate model-acquisition function based learning framework, where an estimate for the black-box function $\hat{f}$ (i.e. the surrogate model)

is updated sequentially via black-box function evaluations observed until time step $t$. The selection of candidate points for black-box function evaluation is carried out via an acquisition function, which uses the surrogate model $\hat{f}$ as an inexpensive proxy (to make many internal calls) for the black-box function and produces the next candidate point to be evaluated. The sequence proceeds as follows:

Surrogate model updated on $(x_t, f(x_t)) \rightarrow$ Acquisition function makes (many)

calls to Surrogate model to propose $x_{t+1} \rightarrow$ Surrogate model updated on $(x_{t+1}, f(x_{t+1}))$

In the sequel, we propose two representations that can be used as surrogate models for black-box combinatorial functions over categorical variables. These representations serve as direct generalizations of the Boolean surrogate model based on Fourier expansion proposed in Dadkhahi et al. (2020) in the sense that they reduce to the Fourier representation for real-valued Boolean functions when the cardinality of the categorical variables is two. In addition, both approaches can be modified to address the more general case where different variables are of different cardinalities. However, for ease of exposition, we assume that all the variables are of the same cardinality. Finally, we introduce two popular acquisition function to be used in conjunction with the proposed surrogate models in order to propose new queries for subsequent black-box function evaluations.

## 3.1 REPRESENTATIONS FOR THE SURROGATE MODEL

We present two representations for combinatorial functions $f : [k]^n \rightarrow \mathbb{R}$ and an algorithm to update from the black-box evaluations. The representations use the Fourier basis in two different ways.

**Abridged One-Hot Encoded Boolean Fourier Representation**: The one-hot encoding of each variable $x_i : i \in [n]$ can be expressed as a $(k-1)$-tuple $(x_{i1}, x_{i2}, \ldots, x_{i(k-1)})$, where $x_{ij} \in \{-1, 1\}$ are Boolean variables with the constraint that at most one such variable can be equal to $-1$ for any given $x_i \in [k]$.

We consider the following representation for the combinatorial function $f$:

$$f_\alpha(x) = \sum_{m=0}^{n} \sum_{\mathcal{I} \in \binom{[n]}{m}} \sum_{\mathcal{J} \in [k-1]^{|\mathcal{I}|}} \alpha_{\mathcal{I},\mathcal{J}} \psi_{\mathcal{I},\mathcal{J}}(x) \tag{2}$$

where $[k-1]^{|\mathcal{I}|}$ denotes $|\mathcal{I}|$-fold cartesian product of the set $[k-1] = \{1, 2, \ldots, k-1\}$, $\binom{[n]}{m}$ designates the set of $m$-subsets of the set $[n]$, and the monomials $\psi_{\mathcal{I},\mathcal{J}}(x)$ can be written as

$$\psi_{\mathcal{I},\mathcal{J}}(x) = \prod_{\{(i,j):i=\mathcal{I}_\ell, j=\mathcal{J}_\ell, \ell \in [|\mathcal{I}|]\}} x_{ij} \tag{3}$$

A second order approximation (i.e. at $m = 2$) of the representation in (2) can be expanded in the following way:

$$\widehat{f}_\alpha(x) = \alpha_0 + \sum_{i \in [n]} \sum_{\ell \in [k-1]} \alpha_{i\ell} x_{i\ell} + \sum_{(i,j) \in \binom{[n]}{2}} \sum_{(p,q) \in [k-1]^2} \alpha_{ijpq} x_{ip} x_{jq}. \tag{4}$$

**Example.** *For $n = 2$ variables $x_1$ and $x_2$, each of which with cardinality $k = 3$, we have the one-hot encoding of $(x_{11}, x_{12})$ and $(x_{21}, x_{22})$ respectively. From Equation (4), the one-hot encoding factorization for this example can be written as*

$$f(x) = \alpha_0 + \alpha_1 x_{11} + \alpha_2 x_{12} + \alpha_3 x_{21} + \alpha_4 x_{22} + \alpha_5 x_{11} x_{21} + \alpha_6 x_{11} x_{22} + \alpha_7 x_{12} x_{21} + \alpha_8 x_{12} x_{22}.$$

Note that the representation in Equation (2) has far less terms than a vanilla one-hot encoding with all the combinations of one-hot variables included (as suggested in Ricardo Baptista (2018)). The reason for this reduction is two-fold: $(i)$ $(k-1)$ Boolean variables model each categorical variable of cardinality $k$, and more importantly $(ii)$ each monomial term has at most one Boolean variable $x_{ij}$ from its corresponding parent categorical variable $x_i$. The following theorem states that this reduced representation is in fact unique and complete.

**Theorem 3.1.** *The representation in Equation (2) is complete and unique for any real-valued combinatorial function.*

*Proof.* See Appendix. □

**Fourier Representation on Finite Abelian Groups**: We define a cyclic group structure $\mathbb{Z}/k_i\mathbb{Z}$ over the elements of each categorical variable $x_i$ ($i \in [n]$), where $k_i$ is the cardinality of the latter variable. From the fundamental theorem of abelian groups Terras (1999), there exists an abelian group $G$ which is isomorphic to the direct sum (a.k.a direct product) of the cyclic groups $\mathbb{Z}/k_i\mathbb{Z}$ corresponding to the $n$ categorical variables:

$$G \cong \mathbb{Z}/k_1\mathbb{Z} \oplus \mathbb{Z}/k_2\mathbb{Z} \oplus \ldots \oplus \mathbb{Z}/k_n\mathbb{Z} \tag{5}$$

where the latter group consists of all vectors $(a_1, a_2, \ldots, a_n)$ such that $a_i \in \mathbb{Z}/k_i\mathbb{Z}$ and $\cong$ denotes group isomorphism. We assume that $k_i = k$ ($\forall i \in [n]$) for simplicity, but the following representation could be easily generalized to the case of arbitrary cardinalities for different variables.

The Fourier representation of any complex-valued function $f(x)$ on the finite abelian group $G$ is given by Terras (1999)

$$f(x) = \sum_{\mathcal{I} \in [k]^n} \alpha_{\mathcal{I}} \psi_{\mathcal{I}}(x) \tag{6}$$

where $\alpha_{\mathcal{I}}$ are (in general complex) Fourier coefficients, $[k]^n$ is the $n$-fold cartesian product of the set $[k]$ and $\psi_{\mathcal{I}}(x)$ are complex exponentials [1] ($k$-th roots of unity) given by

$$\psi_{\mathcal{I}}(x) = \exp\left(2\pi j \langle x, \mathcal{I} \rangle / k\right).$$

Note that the latter complex exponentials are the *characters* of the representation, and reduce to the *monomials* (i.e. in $\{-1, 1\}$) when the cardinality of each variable is two. A second order approximation of the representation in (6) can be written as:

$$\widehat{f_{\alpha}}(x) = \alpha_0 + \sum_{i \in [n]} \sum_{\ell \in [k-1]} \alpha_{i\ell} \exp\left(2\pi j x_i \ell / k\right) + \sum_{(i,j) \in \binom{[n]}{2}} \sum_{(p,q) \in [k-1]^2} \alpha_{ijpq} \exp\left(2\pi j (x_i p + x_j q) / k\right). \tag{7}$$

For a real-valued function $f_{\alpha}(x)$ (which is of interest here), the representation in (6) reduces to

$$f_{\alpha}(x) = \Re\left\{ \sum_{\mathcal{I} \in [k]^n} \alpha_{\mathcal{I}} \psi_{\mathcal{I}}(x) \right\} = \sum_{\mathcal{I} \in [k]^n} \alpha_{r,\mathcal{I}} \psi_{r,\mathcal{I}}(x) - \sum_{\mathcal{I} \in [k]^n} \alpha_{i,\mathcal{I}} \psi_{i,\mathcal{I}}(x) \tag{8}$$

where

$$\psi_{r,\mathcal{I}}(x) = \cos\left(2\pi \langle x, \mathcal{I} \rangle / k\right) \quad \text{and} \quad \psi_{i,\mathcal{I}}(x) = \sin\left(2\pi \langle x, \mathcal{I} \rangle / k\right) \tag{9}$$

$$\alpha_{r,\mathcal{I}} = \Re\{\alpha_{\mathcal{I}}\} \quad \text{and} \quad \alpha_{i,\mathcal{I}} = \Im\{\alpha_{\mathcal{I}}\} \tag{10}$$

We note that the number of characters utilized in this representation is almost twice as many as that of monomials used in the previous representation.

**Surrogate Model Learning**: We adopt the learning algorithm of combinatorial optimization with expert advice Dadkhahi et al. (2020) in the following way. We consider the monomials $\psi_{\mathcal{I},\mathcal{J}}(x)$ in (3) and the characters $\psi_{\ell,\mathcal{I}}(x)$ in (10) as experts. For each surrogate model, we maintain a pool of such experts, the coefficients of which are refreshed sequentially via an exponential weight update rule. We refer to the proposed algorithm as *Expert-Based Categorical Optimization* (ECO) and the two versions of the algorithm with the two proposed surrogate models are called ECO-F (based on the One-Hot Encoded Boolean Fourier Representation) and ECO-G (based on Fourier Representation on Finite Abelian Groups), respectively. A summary of the algorithm is given in the Appendix.

### 3.2 ACQUISITION FUNCTIONS

In this subsection, we discuss how two popular acquisition functions, namely Simulated Annealing (SA) and Monte Carlo Tree Search (MCTS), work with a surrogate model and use cost-free evaluations of the surrogate model to select the next query for the black box evaluation. In the literature, SA has been used for the generic BBO problems whereas MCTS has been used for the design problems.

---

[1]Note that in the general case of different cardinalities for different variables, $\mathcal{I} \in [k_1] \times [k_2] \times \ldots \times [k_n]$ where $\times$ denotes the cartesian product and the exponent denominator in the complex exponential character is replaced by $k = \mathsf{LCM}(k_1, k_2, \ldots, k_n)$.

**SA as Acquisition Function:** Our acquisition function is devised so as to minimize $\widehat{f}_\alpha(x)$, the current estimate for the surrogate model. A simple strategy to minimize $\widehat{f}_\alpha(x)$ is to iteratively switch each variable into the value that minimizes $\widehat{f}_\alpha$ given the values of all the remaining variables, until no more changes occur after a sweep through all the variables $x_i$ ($\forall i \in [n]$). A strategy to escape local minima in this context Pincus (1970) is to allow for occasional increases in $\widehat{f}_\alpha$ by generating a Markov Chain (MC) sample sequence (for $x$), whose stationary distribution is proportional to $\exp(-\widehat{f}_\alpha(x)/s)$, where $s$ is gradually reduced to zero. This optimization strategy was first applied by Kirkpatrick et al. (1983) in their Simulated Annealing algorithm to solve combinatorial optimization problems. We use the Gibbs sampler Geman & Geman (1984) to generate such a MC by sampling from the full-conditional distribution of the stationary distribution, which in our case is given by the softmax distribution over $\{-\widehat{f}_\alpha(x_i = \ell, x_{-i})/s\}_{\ell \in [k]}$, for each variable $x_i$ conditional on the values of the remaining variables $x_{-i}$. By decreasing $s$ from a high value to a low one, we allow the MC to first search at a coarse level avoiding being trapped in local minima.

Algorithm 1 presents our simulated annealing (SA) version for categorical domains, where $s(t)$ is an annealing schedule, which is a non-increasing function of $t$. We use the annealing schedule suggested in Spears (1993), which follows an exponential decay with parameter $\ell$ given by $s(t) = \exp(-\ell t/n)$. In each step of the algorithm, we pick a variable $x_i$ ($i \in [n]$) uniformly at random, evaluate the surrogate model (possibly in parallel) $k$ times, once for each categorical value $\ell \in [k]$ for the chosen variable $x_i$ given the current values $x_{-i}$ for the remaining variables. We then update $x_i$ with the sampled value in $[k]$ from the corresponding softmax distribution.

**MCTS as Acquisition Function:** We formulate the design problem as an undiscounted Markov decision process $(\mathcal{S}, \mathcal{A}, T, R)$. Each state $s \in \mathcal{S}$ corresponds to a partial or full sequence of categorical variables of lengths in $[0, n]$. The process in each episode starts with an empty sequence $s_0$, the initial state. Actions are selected from the set of permissible additions to the current state (sequence) $s_t$ at each time step $t$, $\mathcal{A}_t = \mathcal{A}(s_t) \subset \mathcal{A}$. The transition function $T$ is deterministic, and defines the sequence obtained from the juxtaposition of the current state $s_t$ with the action $a_t$, i.e. $s_{t+1} = T(s_t, a_t) = s_t \circ a_t$. The transitions leading to incomplete sequences yield a reward of zero. Complete sequences are considered as terminal states, from which no further transitions (juxtapositions) can be made. Once the sequence is complete (i.e. at a terminal state), the reward is obtained from the current surrogate reward model $\widehat{f}_\alpha$. Thus, the reward function is defined as $R(s_t, a_t, s_{t+1}) = -\widehat{f}_\alpha(s_{t+1})$ if $s_{t+1}$ is terminal, and zero otherwise.

MCTS is a popular search algorithm used for design problems. MCTS is a rollout algorithm which keeps track of the value estimates obtained via Monte Carlo simulations in order to progressively make better selections. The UCT selection criteria, see Kocsis & Szepesvári (2006), is typically used as tree policy, where action $a_t$ at state $s_t$ in the tree search is selected via: $\pi^{\mathcal{T}}(s_t) = \arg\max_{a \in \mathcal{A}(s_t)} Q(s_t, a) + c\sqrt{\ln N(s_t)/N(s_t, a)}$, where $\mathcal{T}$ is the search tree, $c$ is the exploration parameter, $Q(s, a)$ is the state-action value estimate, and $N(s)$ and $N(s, a)$ are the visit counts for the parent state node and the candidate state-action edge, respectively. For the selection of actions in states outside the tree search, a random default policy is used: $\pi^{RS}(s_t) = \mathtt{unif}(\mathcal{A}_t)$.

A summary of the proposed algorithm is given in Algorithm 2. Starting with an empty sequence $s_0$ at the root of the tree, we follow the tree policy until a leaf node of the search tree is reached (selection step). At this point, we append the state corresponding to the leaf node to the tree and initialize a value function estimate for its children (extension step). From the reached leaf node we follow the default policy until a terminal state is reached. At this point, we plug the sequence corresponding to this terminal state into the surrogate reward model $-\widehat{f}_\alpha$ and observe the reward $r$. This reward is backed up from the leaf node to the root of the tree in order to update the value estimates $Q(s, a)$ via Monte Carlo (i.e. using the average reward) for all visited $(s, a)$ pairs along the path. This process is repeated until a stopping criterion (typically a max number of playouts) is met, at which point the sequence $s_{\mathtt{best}}$ with maximum reward $r_{\mathtt{best}}$ is returned as the output of the algorithm.

**Algorithm 1** SA for Categorical Variables with Surrogate Model

1: **Inputs:** surrogate model $\widehat{f}_\alpha$, annealing schedule $s(t)$, categorical domain $\mathcal{X}$
2: Initialize $x \in \mathcal{X}$
3: $t = 0$
4: **repeat**
5:    $i \sim \mathtt{unif}([n])$
6:    $x_i | x_{-i}$                       $\sim$
     $\mathtt{Softmax}\big(\{-\widehat{f}_{\alpha_t}(x_i=\ell,x_{-i})/s(t)\}_{\ell \in [k]}\big)$
7:    $t \leftarrow t+1$
8: **until** Stopping Criteria
9: **return** $x$

**Algorithm 2** MCTS with Surrogate Reward

1: **Inputs:** surrogate model $\widehat{f}_\alpha$, search tree $\mathcal{T}$
2: Initialize $s_{\text{best}} = \{\}, r_{\text{best}} = -\infty$
3: **repeat**
4:    $s_{\text{leaf}} \leftarrow \mathtt{Selection}(\pi^{\mathcal{T}})$
5:    $\mathcal{T} \leftarrow \mathcal{T} \cup \{s_{\text{leaf}}\}$
6:    $s_t \leftarrow \mathtt{Simulation}(\pi^{RS}, s_{\text{leaf}})$
7:    $r \leftarrow -\widehat{f}_\alpha(s_t)$
8:    $\mathtt{Backup}(s_{\text{leaf}}, r)$
9:    **if** $r > r_{\text{best}}$ **then**
10:      $r_{\text{best}} \leftarrow r$ and $s_{\text{best}} \leftarrow s_t$
11:    **end if**
12: **until** Stopping Criteria
13: **return** $s_{\text{best}}$

**Computational Complexity**: The computational complexity per time step associated with learning the surrogate model, for both representations introduced in 3.1, is in $\mathcal{O}(d) = \mathcal{O}(k^{m-1}n^m)$, and is thus linear in the number of experts $d$. Moreover, the complexity of the simulated annealing algorithm (Algorithm 1) is in $\mathcal{O}(kk^{m-1}n^{m-1}n) = \mathcal{O}(kd)$, assuming that the number of iterations in each SA run is in $\mathcal{O}(n)$. As a result, the overall complexity of the algorithm is in $\mathcal{O}(kd)$. Finally, the computational complexity of each playout in Algorithm 2 is in $(O)(kn)$, leading to an overall complexity of $(O)(kd)$, assuming $(O)(d/n)$ playouts per time step.

## 4 EXPERIMENTS AND RESULTS

In this section, we measure the performance of the proposed representations, when used as surrogate/reward model in conjunction with search algorithms (SA and MCTS) in BBO and design problems. The learning rate used in exponential weight updates is selected via the anytime learning rate schedule suggested in Dadkhahi et al. (2020) and Gerchinovitz & Yu (2011) (see Appendix). The maximum degree of interactions used in our surrogate models is set to two for all the problems; increasing the max order improved the results only marginally. The sparsity parameter $\lambda$ in exponential weight updates is set to 1 in all the experiments following the same choice made in Dadkhahi et al. (2020). Experimentally, the learning algorithm is fairly insensitive to the variations in the latter parameter. In each experiment, we report the results averaged over multiple runs (20 runs in BBO experiments and 10 runs in design experiments) $\pm$ one standard error of the mean. The experiments were run on machines with CPU cores from the Intel Xeon E5-2600 v3 family.

**BBO Experiments**: We compare the performance of our ECO algorithms in conjunction with SA with two baselines, random search (RS) and simulated annealing (SA), as well as a state-of-the-art Bayesian combinatorial optimization algorithm (COMBO) Oh et al. (2019). In particular, we consider two synthetic benchmarks (Latin square problem and pest control problem) and a real-word sequence design problem in biology: RNA sequence optimization. In addition to the performance of the algorithms in terms of the best value of $f(x)$ observed until a given time step $t$, we measure the average computation time per time step of our algorithm versus that of COMBO. The decay parameter used in the annealing schedule of SA is set to $\ell = 3$ in all the experiments. In addition, the number of SA iterations in our surrogate models is set to $T = 3 \times n$. Intuitively, each of these parameters creates an exploration-exploitation trade-off. The smaller (larger) the value of $\ell$ or $T$, the more exploratory (exploitative) is the behavior of SA. The selected values seem to create a reasonable balance; tuning these parameters may improve the performance of the acquisition function.

**Synthetic Benchmarks**: We consider two synthetic problems: Latin square problem Colbourn & Dinitz (2006), a commonly used combinatorial optimization benchmark, and the pest control problem considered in Oh et al. (2019) (see Appendix for the latter results). In both problems, we have $n = 25$ categorical variables, each of cardinality $k = 5$. A Latin square of order $k$ is a $k \times k$ matrix of elements $x_{ij} \in [k]$, such that each number appears in each row and column exactly once. When $k = 5$, the problem of finding a Latin square has $161,280$ solutions in a space of dimensionality $5^{25}$. We formulate the problem of finding a Latin square of order $k$ as a black-box optimization by

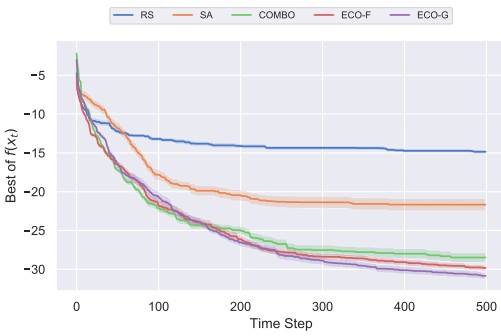 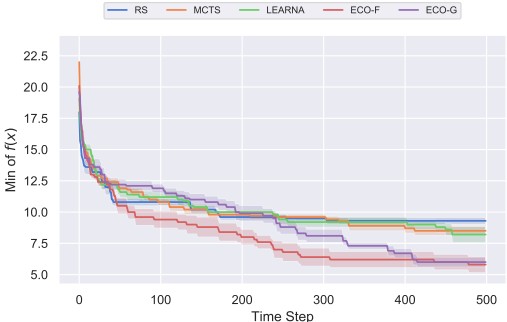

Figure 1: RNA BBO Problem with $n = 30$      Figure 2: Eterna100 puzzle #41 ($n = 35$)

imposing an additive penalty of one for any repetition of numbers in any row or column. Hence, function evaluations are in the range $[0, 2k(k-1)]$, and a function evaluation of zero corresponds to a Latin square of order $k$. We consider a noisy version of this problem, where an additive Gaussian noise with zero mean and standard deviation of $0.1$ is added to function evaluations observed by each algorithm. Both ECO-F and ECO-G outperform the baselines with a considerable margin. In addition, ECO-G outperforms COMBO until time step $t = 190$. At larger time steps, COMBO outperforms the other algorithms, however, this performance comes at the price of a far larger computation time. ECO-F and ECO-G offer a speed-up over COMBO by a factor of roughly $100$ and $50$, respectively.

**RNA Sequence Optimization Problem**: Consider an RNA sequence as a string $A = a_1 \ldots a_n$ of $n$ letters (nucleotides) over the alphabet $\Sigma = \{A, U, G, C\}$. A pair of complementary nucleotides $a_i$ and $a_j$, where $(i < j)$, can interact with each other and form a base pair (denoted by $(i, j)$), A-U, C-G and G-U being the energetically stable pairs. Thus, the secondary structure, i.e. the minimum free energy structure, of an RNA can be represented by an ensemble of pairing bases. A number of RNA folding algorithms Lorenz et al. (2011); Markham & Zuker (2008) use a thermodynamic model (e.g. Zuker & Stiegler (1981)) and dynamic programming to estimate MFE of a sequence. However, the $O(n^3)$ time complexity of these algorithms prohibits their use for evaluating substantial numbers of RNA sequences Gould et al. (2014) and exhaustively searching the space to identify the global free energy minimum, as the number of sequences grows exponentially as $4^n$.

We formulate the RNA sequence optimization problem as follows: For a sequence of length $n$, find the RNA sequence which folds into a secondary structure with the lowest MFE. In our experiments, we initially set $n = 30$ and $k = 4$. We then use the popular RNAfold package Lorenz et al. (2011) to evaluate the MFE for a given sequence. The goal is to find the lowest MFE sequence by calling the MFE evaluator minimum number of times. As depicted in Figure 1, both ECO-F and particularly ECO-G outperform the baselines as well as COMBO by a considerable margin.

**RNA Design Experiments**: The problem is to find a primary RNA sequence $\phi$ which folds into a target structure $w$, given a folding algorithm $F$. Such target structures can be represented as a sequence of dots (for unpaired bases) and brackets (for paired bases). In our algorithm, the action sets are defined as follows. For unpaired sites $\mathcal{A}_t = \{A, G, C, U\}$ and for paired sites $\mathcal{A}_t = \{GC, CG, AU, UA\}$. At the beginning of each run of our algorithm (ECO-F and ECO-G in conjunction with MCTS acquisition), we draw a random permutation for the order of locations to be selected in each level of the tree. The reward value offered by the environment (i.e. the black-box function) at any time step $t$ corresponds to the normalized Hamming distance between the target structure $\omega$ and the structure $y_t = F(x_t)$ of the sequence $x_t$ found by each algorithm, i.e. $d_H(w, y_t)$.

We compare the performance of our algorithms against RS as a baseline, where random search is carried out over the given structure (i.e. default policy $\pi^{\mathrm{RS}}$) rather than over unstructured random sequences. We also include two state-of-the-art algorithms in our experiments: MCTS-RNA of Yang et al. (2017) and LEARNA of Runge et al. (2019). MCTS-RNA has an exploration parameter, which we tune in advance (per sequence). LEARNA has a set of $14$ hyper-parameters tuned a priori using training data and is provided by the authors of Runge et al. (2019). Note that the latter training phase (for LEARNA) as well as the former exploration parameter tuning (for MCTS-RNA) is offered to the respective algorithms as an advantage, whereas for our algorithm we use a global set of heuristic choices for the two hyper-parameters, rather than attempting to tune the two hyper-parameters. In particular, we set the exploration parameter $c$ to $0.5$ and the number of MCTS playouts at each time

step to $30 \times h$, where $h$ is the height of the tree (i.e. number of dots and bracket pairs). The latter heuristic choice is made since the bigger the tree, the more playouts are needed to explore the space.

We point out that the entire design pipeline in state-of-the-art algorithms typically also includes a local improvement step (as a post-processing step), which is either a rule-based search (e.g. in Yang et al. (2017)) or an exhaustive search (e.g. in Runge et al. (2019)) over the mismatched sites. We do not include the local improvement step in our experiments, since we are interested in measuring sample efficiency of different algorithms. In other words, the question is the following: given a fixed and finite evaluation budget, which algorithm is able to get closer to the target structure.

For our experiments, we focus on three puzzles from the Eterna-100 dataset Anderson-Lee et al. (2016). Two of the selected puzzles (#15 and #41 of lengths 30 and 35, resp.), despite their fairly small lengths, are challenging for many algorithms (see Anderson-Lee et al. (2016)). In both puzzles, our algorithms ECO-F and ECO-G (with MCTS acquisition) are able to significantly improve the performance of MCTS when limited number of black-box evaluations is available. All algorithms outperformed RS as expected. Within the given $500$ evaluation budget, ECO-G, and especially ECO-F, are superior to LEARNA by a substantial margin (see Appendix). In puzzle number $41$ (Figure 2), again both ECO-G and ECO-F significantly outperform LEARNA, over the given number of evaluations. Interestingly, ECO-F is able to outperform LEARNA throughout the evaluation process, and in average finds a far better final solution than LEARNA. See Appendix for the third puzzle.

## 5 CONCLUSIONS AND FUTURE WORK

In summary, we propose novel Fourier representations as surrogate models for black box optimization over categorical variables and show performance improvements over existing baselines when combined with state-of-the-art acquisition methods.

Considering the performance variability of the two surrogate model representations introduced in this paper across different problems, an important research avenue would be to incorporate an ensemble of surrogate models rather than a single one. Such an ensemble model would then update and explore both models simultaneously and draw samples from either individual models or a combination of both at any given time step. It would be interesting to see if such an ensemble model would in fact be able to outperform both individual models over different combinatorial problems.

Our ECO algorithm incorporates an online estimator learnt via Hedge, rather than a Bayesian posterior mean function which is commonly used in conjunction with Thompson Sampling (TS) and UCB for uncertainty quantification in Bayesian optimization algorithms. The Hedge algorithm has strong adversarial guarantees (see Dadkhahi et al. (2020) for theoretical results in the Boolean case), which can be shown to carry over to our setting as well. More precisely, given any additional black box evaluation, it is guaranteed to move closer to the true black-box model. However, the exploration bonus used in the acquisition process via SA, used in our algorithm, can be shown to be *domain independent*, i.e. sampled i.i.d from the same distribution regardless of the query point. The terms that account for uncertainty in both TS and UCB are domain dependent and depend on the query point. As such, domain dependent uncertainty incorporation would be an interesting next step, which is left for future work.

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

## A   PROOF OF THEOREM 3.1

We first assume that the one-hot variables $x_{ij} \in \{0, 1\}$. Plugging different choices of $x \in \mathcal{X} = [k]^n$ into Equation (2) leads to a system of linear equations with $k^n$ unknowns (coefficients $\alpha_{\mathcal{I},\mathcal{J}}$) and $k^n$ equations. We can express this system in matrix form as the product of the matrix of monomials (where each column $j$ corresponds to a monomial $\psi_j$, and each element $(i, j)$ corresponds to the evaluation of the monomial $\psi_j$ at the $i$-th choice for $x$) and the vector of unknown coefficients, which is set equal to the function values at the corresponding choices for $x$. We claim that there exists a permutation of the rows and columns of the matrix of monomials such that the latter becomes unit lower triangular, and is thereby full rank. As a result, the representation in (2) for $x_{ij} \in \{0, 1\}$ is complete and unique.

To formally show that this permutation for the monomials' matrix exists, we use a construction by induction over the number of variables $n$ included in the representation. We denote the monomials' matrix over $\ell$ variables with $\Phi_\ell$, and define the one-hot variables $x_{ij} (j \in [k-1])$ as the *descendants* of the *parent* categorical variable $x_i$ ($\forall i \in [n]$). It is easy to see that such a construction for the base case of only one variable exits, as the monomials' matrix is a $k \times k$ matrix. In this case, we use the following permutation of the monomials (columns): $(1, x_{11}, x_{12}, \ldots, x_{1(k-1)})$. We also use the following permutation of the $x_1$ values in rows: $(k, 1, \ldots, k-1)$. As a result, in this matrix only the elements of the first column and the main diagonal are non-zero and equal to one, and thus the matrix $\Phi_1$ is unit lower triangular.

Assuming that the induction hypothesis holds for $n$ variables, we show that it also holds for $n+1$ variables. Starting from a unit lower triangular matrix $\Phi_n$, we can construct the matrix $\Phi_{n+1}$ as follows. Note that all the $k^n$ columns of $\Phi_n$ correspond to the monomials composed of the descendants of the first $n$ variables $\{x_{ij} : \forall i \in [n] \text{ and } \forall j \in [k-1]\}$, whereas each of the additional $k^{n+1} - k^n$ columns introduced in $\Phi_{n+1}$ involves exactly one term from $\{x_{(n+1)j} : \forall j \in [k-1]\}$ (possibly also containing factors from the previous $n$ variables). We can express $k^{n+1} - k^n$ using the following binomial expansion:

$$k^{n+1} - k^n = (k-1) \sum_{m=1}^{n+1} \binom{n}{m-1} (k-1)^{m-1}. \tag{11}$$

In words, the additional $k^{n+1} - k^n$ columns can be considered as a collection of $\binom{n}{m-1}(k-1)^{m-1}$ $m$-th order monomials ($m \in [n+1]$), each of which includes one out of the $k-1$ descendant variables $x_{(n+1)j}$ ($j \in [k-1]$) together with $m-1$ variables from the (descendants) of the previous $n$ variables. Each of the latter $m-1$ variables can take values in $[k-1]$, whereas the remaining $n - m + 1$ variables are set to $k$.

Starting with $m = 1$, we have $(k-1)$ first order terms $(x_{(n+1)1}, x_{(n+1)2}, \ldots, x_{(n+1)(k-1)})$ which we assign to columns $(k^n+1, \ldots, k^n+(k-1))$. The $x$ values associated with rows $(k^n+1, \ldots, k^n+(k-1))$ are constructed by assuming that $(i)$ $x_{n+1}$ takes values from 1 to $k-1$ (in order), while $(ii)$ all the remaining $x_j$ ($j \in [n]$) variables are set to $k$. As a consequence of $(i)$, all the elements on the main diagonal are ones; also, as a consequence of $(ii)$, all the higher degree monomials involving $x_{n+1}$ (which occupy the elements after the diagonal ones) are equal to zeros. Thus the augmented matrix, until this point, remains unit lower triangular.

We then consider the second degree terms $m = 2$, where we have $n(k-1)$ terms (choice of one out of the other $n$ variables, each taking values from $[k-1]$) for each of the $k-1$ choices for the variable $x_{n+1}$. Starting with second order monomials involving $x_1$ and $x_{n+1}$, we again assume that all the remaining variables $x_i$ ($i \notin \{1, n+1\}$) are equal[2] to $k$. For any choice of $(x_1, x_{n+1}) \in [k-1]^2$, we add a new column corresponding to the monomial $x_{1j}x_{(n+1)\ell}$ as well as a new row in which $x_i = j$ and $x_{n+1} = \ell$, whereas the remaining variables are set to $k$. As a result, we have that: $(i)$ the diagonal element in the new row/column is equal to one, and $(ii)$ all the elements in the future[3] columns are zeros, since any combination of one of the descendants of $x_{n+1}$ with the remaining

---

[2] Note that this assumption is necessary in order to ensure that monomials in future columns for the same row are evaluated to zero; choices of $x$ where this assumption is not valid is addressed in next rows.

[3] Note that the elements in the previous columns in the same row corresponding to monomials involving the selected $m-1$ variables are non-zeros as well.

variables (any variable except $x_1$ and $x_{n+1}$) is zero. We continue this construction strategy for the remaining variables until all the second degree terms involving $x_{n+1}$ and one out of the remaining $n-1$ variables is exhausted. We then repeat the same idea for terms with orders up to $n+1$, as defined by the binomial expansion in (11). As a result of this construction strategy, the monomial matrix $\Phi_{n+1}$ is unit lower triangular.

Now, we use this result to show the completeness and uniqueness of the representation with one-hot variables in $\{-1, 1\}$ in the following way.

**Completeness**: We showed that the representation (2) over $\{0, 1\}$ is complete, i.e. we can express any function using the representation in (2), where we have at most one descendant term from the same parent in each monomial. Now, we replace each $x_{ij}$ (from $\{0, 1\}$) in the latter representation with $(1-x'_{ij})/2$, where $x'_{ij} \in \{1, -1\}$. The new representation can also be expressed via the expansion (2) since no two descendants from the same parent variable are being multiplied with each another.

**Uniqueness**: Assume that the uniqueness condition is not satisfied. Then we have two distinct polynomial representations $f_1$ and $f_2$ that have the same value for every $x$. However, since $f_1$ and $f_2$ are distinct polynomials, $f_1(x) - f_2(x)$ is a polynomial $p(x)$ which is non-zero in at least one input $x^*$. This implies that $f_1(x^*) - f_2(x^*)$ is also non-zero, which is a contradiction, and the proof is complete.

**Remark 1.** *The group-theoretic Fourier representation defined in Equation* (6) *is unique and complete.*

*Proof.* Let $\chi = [k]^n$ be the categorical domain. Let the true function be $f$. For generality, let us consider a complex valued function $f : \chi \to \mathbb{C}$ where $\mathbb{C}$ is the field of complex numbers. The basis functions are $\psi_{\mathcal{I}}(x)$. Now, one can view a function as a $[k]^n$-length vector, one entry each for evaluating the function at every point in the domain $\chi$. We denote the vector for function $f$, thus obtained, by $f^\chi \in \mathbb{C}^{k^n}$. Similarly, denote the vector for evaluations of the basis function $\psi_{\mathcal{I}}$ by $\psi_{\mathcal{I}}^\chi \in \mathbb{C}^{k^n}$. Let $A$ be a matrix created by stacking all vectors corresponding to basis vectors in the columns. Then, the Fourier representation can be written as $f^\chi = A\alpha$ where $\alpha$ is the vector of Fourier coefficients in our group-theoretic representation. Now, due to the use of complex exponentials, one can show that $\sum_{x \in [k]^n} \psi_{\mathcal{I}}(x)\psi_{\mathcal{I}'}(x) = 0$ if $\mathcal{I} \neq \mathcal{I}'$. Therefore, the columns of the matrix $A$ are orthogonal. Hence, $A$ is a full rank matrix. Therefore, our representation is merely representing a vector in another full rank orthogonal basis. Hence, it is unique and complete. $\qquad\square$

# B    DESCRIPTION OF ALGORITHMS

**Surrogate Model Learning Algorithm**: Let $n$ and $k$ denote the number of variables and the cardinality of each variable, respectively. The surrogate models used in ECO-F and ECO-G correspond to approximations of the representations given in (2) and (8), respectively, where each approximation is obtained by restricting the maximum order of interactions among variables to $m$. We consider each term in the latter surrogate models, i.e. monomials $\psi_{\mathcal{I},\mathcal{J}}$ from (3) in ECO-F and characters $\psi_{\beta,\mathcal{I}}(\beta \in \{r, i\})$ from (10) in ECO-G, as an expert, denoted by $\psi_i$ ($i \in [d]$). The number of such experts in ECO-F is $d = \sum_{i=0}^m \binom{n}{i}(k-1)^i$ which coincides with the dimensionality of the space $k^n$ when $m = n$, whereas the number of experts in ECO-G is equal to $d = 2\sum_{i=0}^m \binom{n}{i}(k-1)^i - 1$. The coefficient of each expert $\psi_i$ is designated by $\alpha_i$. Since the exponential weights, utilized to update the coefficients $\alpha_i$, are non-negative, we maintain two non-negative coefficients $\alpha_i^+$ and $\alpha_i^-$, which yield $\alpha_i = \alpha_i^+ - \alpha_i^-$.

We initialize all the coefficients with a uniform prior, i.e. $\alpha_i^\gamma = 1/2d$ ($\forall i \in [d]$ and $\gamma \in \{-, +\}$). In each time step $t$, we draw a sample $x_t$ via Algorithm 1 with respect to our current estimate for the surrogate model $\widehat{f_\alpha}$. The latter sample is then plugged into the black-box function to obtain the evaluation $f(x_t)$. This leads to a mixture loss $\ell^t$ as the difference between the evaluations obtained by our surrogate model and the black-box function for query $x_t$. Using this mixture loss, we compute the individual loss $\ell_i^t$ for each expert $\psi_i$. Finally, we update each coefficient in the model via an exponential weight obtained according to its incurred individual loss. We repeat this process until stopping criteria are met.

---

**Algorithm 3** Expert Categorical Optimization

---

1: **Inputs:** sparsity $\lambda$, max model order $m$
2: $t \leftarrow 0, \forall \gamma \in \{-,+\} \ \forall i \in [d] : \alpha_{i,\gamma}^t \leftarrow \frac{1}{2d}$
3: **repeat**
4:      $x_t \sim \widehat{f}_{\alpha^t}$ via Algorithm 1 or Algorithm 2
5:      Observe $f(x_t)$
6:      $\widehat{f}_{\alpha^t}(x) \leftarrow \sum_{i \in [d]} \left( \alpha_{i,+}^t - \alpha_{i,-}^t \right) \psi_i(x)$
7:      $\ell^{t+1} \leftarrow \widehat{f}_{\alpha^t}(x_t) - f(x_t)$
8:      **for** $i \in [d]$ and $\gamma \in \{-,+\}$ **do**
9:          $\ell_i^{t+1} \leftarrow 2 \lambda \ell^{t+1} \psi_i(x_t)$
10:         $\alpha_{i,\gamma}^{t+1} \leftarrow \alpha_{i,\gamma}^t \exp\left( - \gamma \eta_t \ell_i^{t+1} \right)$
11:         $\alpha_{i,\gamma}^{t+1} \leftarrow \lambda \cdot \frac{\alpha_{i,\gamma}^{t+1}}{\sum_{\mu \in \{-,+\}} \sum_{j \in [d]} \alpha_{j,\mu}^{t+1}}$
12:      **end for**
13:      $t \leftarrow t + 1$
14: **until** Stopping Criteria
15: **return** $\widehat{x}* = \arg\min_{\{x_i : \forall i \in [t]\}} f(x_i)$

---

**Number of Experts**: The number of terms in vanilla one-hot encoded Fourier representation is $2^{kn}$, whereas our abridged representation reduces this number to $k^n$ matching the space dimensionality, thereby making the algorithm computationally tractable and efficient. When a max degree of $m$ is used in the approximate representation, the number of terms in the abridged representation is equal to $d = \sum_{i=0}^m \binom{n}{i} (k-1)^i$. The corresponding number in a vanilla one-hot encoded representation is equal to $\sum_{i=0}^m \binom{nk}{i}$. Finally, the numbers of terms in the full and order-$m$ group-theoretic Fourier expansions are equal to $2k^n - 1$ and $d = 2 \sum_{i=0}^m \binom{n}{i}(k-1)^i - 1$, respectively.

**MCTS Algorithm**: For a complete version of Algorithm 2, see Algorithm 4.

**Learning Rate**: The anytime learning rate (at time step $t$) used in Algorithm 3 is given by Gerchinovitz & Yu (2011); Dadkhahi et al. (2020):

$$\eta_t = \min\left\{ \frac{1}{e_{t-1}}, c \sqrt{\frac{\ln(2d)}{v_{t-1}}} \right\}, \tag{12}$$

where $c \triangleq \sqrt{2(\sqrt{2}-1)/(\exp(1)-2)}$ and

$$z_{j,t}^\gamma \triangleq -2 \gamma \lambda \ell_t \psi_j(x_t)$$

$$e_t \triangleq \inf_{k \in \mathbb{Z}} \left\{ 2^k : 2^k \geq \max_{s \in [t]} \max_{\substack{j,k \in [d] \\ \gamma,\mu \in \{-,+\}}} |z_{j,s}^\gamma - z_{k,s}^\mu| \right\}$$

$$v_t \triangleq \sum_{s \in [t]} \sum_{\substack{j \in [d] \\ \gamma \in \{-,+\}}} \alpha_{j,s}^\gamma \left( z_{j,s}^\gamma - \sum_{\substack{k \in [d] \\ \mu \in \{-,+\}}} \alpha_{k,s}^\mu z_{k,s}^\mu \right)^2.$$

## C CONTINUED RELATED WORK

A variety of discrete search algorithms and meta-heuristics have been studied in the literature for combinatorial optimization over categorical variables. Such algorithms, including Genetic Algorithms Holland & Reitman (1978), Simulated Annealing Spears (1993), and Particle Swarms Kennedy & Eberhart (1995), are generally inefficient in finding the global minima. In the context of biological sequence optimization, the most popular method is directed evolution Arnold (1998), which explores the space by only making small mutations to existing sequences. In the context of sequence optimization, a recent promising approach consists of fitting a neural network model to predict the black box function and then applying gradient ascent on the latter model Killoran et al.

---

**Algorithm 4** MCTS with Surrogate Reward Model

---

1: **Inputs:** surrogate reward model $\widehat{f}_\alpha$, exploration parameter $c$, search tree $\mathcal{T}$
2: $s_{\text{best}} = \{\}, r_{\text{best}} = -\infty$
3: **repeat**
4:     Initialize episode $t = 0, s_t = []$
5:     **while** $s_t \notin \mathcal{T}$ **do**
6:         $a_t \leftarrow \pi^{\mathcal{T}}(s_t) = \arg\max_{a \in \mathcal{A}_t} Q(s_t, a) + c\sqrt{\ln N(s_t)/N(s_t, a)}$
7:         $s_{t+1} \leftarrow T(s_t, a_t) = s_t \circ a_t$
8:         $t \leftarrow t + 1$
9:     **end while**
10:     $s_{\text{leaf}} = s_t$
11:     $\mathcal{T} \leftarrow \mathcal{T} \cup \{s_t\}$
12:     $\forall a \in \mathcal{A}_{t+1} : N(s_t, a) = 0, \ Q(s_t, a) = 0$
13:     **repeat**
14:         $a_t \leftarrow \pi^{RS}(s_t)$
15:         $s_{t+1} \leftarrow T(s_t, a_t) = s_t \circ a_t$
16:         $t \leftarrow t + 1$
17:     **until** $s_t$ is terminal
18:     $r \leftarrow -\widehat{f}_\alpha(s_t)$
19:     $s \leftarrow s_{\text{leaf}}$
20:     **repeat**
21:         $N(s, a) \leftarrow N(s, a) + 1$
22:         $Q(s, a) \leftarrow Q(s, a) + \frac{1}{N(s,a)}(r - Q(s, a))$
23:         $s \leftarrow \texttt{parent}(s); a \leftarrow$ visited action on $s$
24:     **until** $s$ is the root node
25:     **if** $r > r_{\text{best}}$ **then**
26:         $r_{\text{best}} \leftarrow r$ and $s_{\text{best}} \leftarrow s_t$
27:     **end if**
28: **until** Stopping Criteria
29: **return** $s_{\text{best}}$

---

(2017); Bogard et al. (2019); Liu et al. (2020). This approach allows for a continuous relaxation of the discrete search space making possible step-wise local improvements to the whole sequence at once based on a gradient direction. However, these methods have been shown to suffer from vanishing gradients Linder & Seelig (2020). Further, the projected sequences in the continuous relaxation space may not be recognized by the predictors, leading to poor convergence. Generative model-based optimization approaches aim to learn distributions whose expectation coincides with evaluations of the black box and try to maximize such expectation Gupta & Zou (2019); Brookes et al. (2019). However, such approaches require a pre-trained generative model for optimization.

## D  BBO EXPERIMENTS

**Latin Square Problem**: A latin square of order $k$ is a $k \times k$ matrix of elements $x_{ij} \in [k]$, such that each number appears in each row and column exactly once. When $k = 5$, the problem of finding a latin square has $161,280$ solutions in a space of dimensionality $5^{25}$. We formulate the problem of finding a latin square of order $k$ as a black-box function by imposing an additive penalty of one for any repetition of numbers in any row or column. As a result, function evaluations are in the range $[0, 2k(k-1)]$, and a function evaluation of zero corresponds to a latin square of order $k$. We consider a noisy version of this problem, where an additive Gaussian noise with zero mean and standard deviation of $\sigma = 0.1$ is added to function evaluations observed by each algorithm.

Figure 3 demonstrates the performance of different algorithms, in terms of the best function value found until time $t$, over 500 time steps. Both ECO-F and ECO-G outperform the baselines with a considerable margin. In addition, both ECO-G and ECO-F match COMBO's performance closely until time step $t = 190$. At larger time steps, COMBO outperforms the other algorithms, however, this performance comes at the price of a far larger computation time. As demonstrated in Table 1, ECO-F and ECO-G offer a speed-up over COMBO by a factor of approximately 100 and 50, respectively.

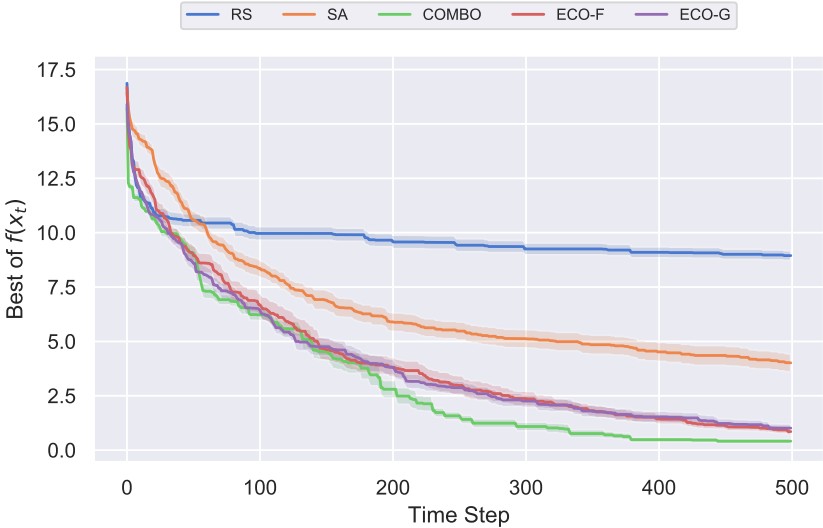

Figure 3: Best function evaluation seen so far for the Latin Square problem.

**Pest Control Problem**: In the pest control problem, given $n$ stations and $k - 1$ pesticide types, the idea is to maintain the spread of pest (with minimum cost), which is propagating throughout the stations in an interactive and probabilistic fashion. The $k$-th category for each variable corresponds to the choice of no pesticide at all. Controlling the spread of the pest is carried out via the choice of the right type of pesticide subject to a penalty proportional to its associated cost. A closed form definition of this problem is given in Oh et al. (2019).

The results for different algorithms are shown in Figure 4. Despite the fact that COMBO is able to find the minimum in fewer time steps (in $\approx 200$ steps) than ECO-F (in $\approx 360$ steps) on average,

ECO-F outperforms COMBO during initial time steps (until $t \approx 180$). SA performs competitively, but eventually is unable to find the optimal solution to this problem over the designated $500$ steps. The poor performance of ECO-G can be explained by the interactive nature of the problem, where early mistakes are punished inordinately. Early mistakes made by ECO-G can also be attributed to the large number of experts (with noisy coefficients) in its model, which in turn promotes an early exploratory behavior.

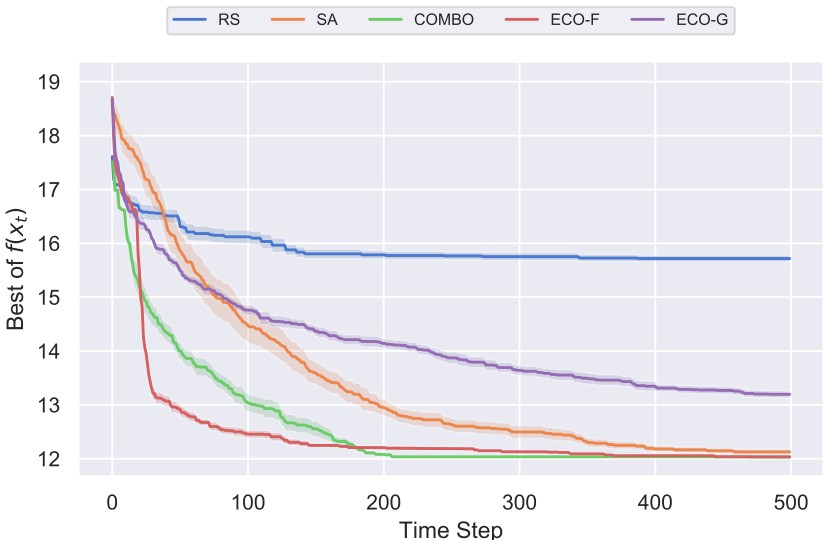

Figure 4: Best function evaluation seen so far for the pest control problem.

**RNA Sequence Optimization Problem**: Structured RNA molecules play a critical role in many biological applications, ranging from control of gene expression to protein translation. The native secondary structure of a RNA molecule is usually the minimum free energy (MFE) structure. Consider an RNA sequence as a string $A = a_1 \ldots a_n$ of $n$ letters (nucleotides) over the alphabet $\Sigma = \{A, U, G, C\}$. A pair of complementary nucleotides $a_i$ and $a_j$, where $(i < j)$, can interact with each other and form a base pair (denoted by $(i, j)$), A-U, C-G and G-U being the energetically stable pairs. Thus, the secondary structure of an RNA can be represented by an ensemble of pairing bases.

Finding the most stable RNA sequences has immediate applications in material and biomedical applications Li et al. (2015). Studies show that by controlling the structure and free energy of a RNA molecule, one may modulate its translation rate and half-life in a cell Buchan & Stansfield (2007); Davis et al. (2008), which is important in the context of viral RNA. A number of RNA folding algorithms Lorenz et al. (2011); Markham & Zuker (2008) use a thermodynamic model (e.g. Zuker & Stiegler (1981)) and dynamic programming to estimate MFE of a sequence. However, the $O(n^3)$ time complexity of these algorithms prohibits their use for evaluating substantial numbers of RNA sequences Gould et al. (2014) and exhaustively searching the space to identify the global free energy minimum, as the number of sequences grows exponentially as $4^n$.

Here, we formulate the RNA sequence optimization problem as follows: For a sequence of length $n$, find the RNA sequence that will fold into the secondary structure with the lowest minimum free energy. In our experiments, we initially set $n = 30$ and $k = 4$. We then use the popular RNAfold package Lorenz et al. (2011) to evaluate the MFE for a given sequence. The goal is to find the lowest MFE sequence by calling the MFE evaluator minimum number of times. The performance of different algorithms is depicted in Figure 1, where both ECO-F and particularly ECO-G outperform the baselines as well as COMBO by a considerable margin.

**Energy-optimized RNA Structures**: Sample RNA sequences obtained via ECO-G after $4000$ time steps for $n = 30$ and $n = 60$ are shown in Figures 6 and 7, respectively. The resulting energy-optimized sequences (as obtained using RNAfold service) have high ($> 90\%$) GC content that makes the strongest positive contribution to lowering MFE Trotta (2014), as pairings between G and C have three hydrogen bonds and are more stable compared to A and U pairings, which

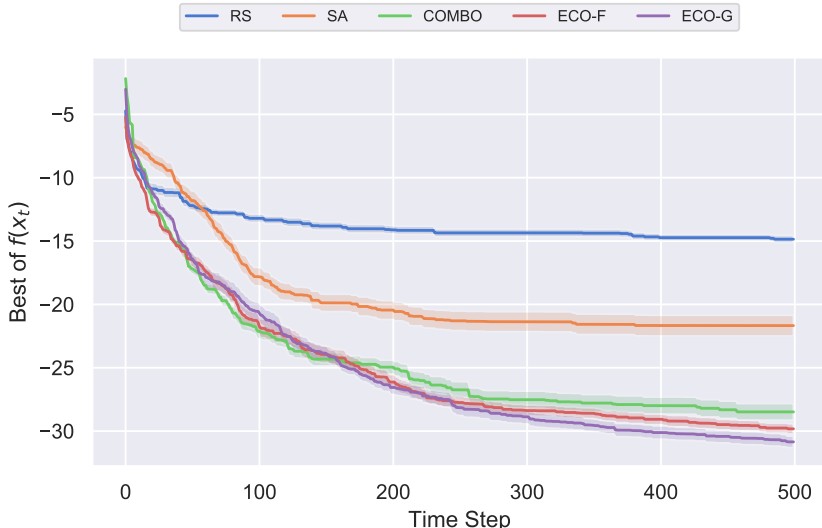

Figure 5: Best function evaluation seen so far for the RNA sequence optimization problem with $n = 30$.

Table 1: Average computation time per step (in Seconds) over different problems and algorithms.

| DATASET | $n$ | $k$ | COMBO | ECO-F | ECO-G |
|---|---|---|---|---|---|
| LATIN SQUARE | 25 | 5 | 170.4 | 1.5 | 3.6 |
| PEST CONTROL | 25 | 5 | 151.0 | 1.4 | 3.3 |
| SEQUENCE PREDICTION | 30 | 4 | 253.8 | 2.0 | 5.7 |

have only two. The final single-strand RNA sequence folds into a GC-paired double helix and a 4 nucleotide long hairpin loop in the middle, which is a tetraloop reported in literature (`http://www.rna.icmb.utexas.edu/SIM/4C/energetics_new/`). Figures 10 and 11 show two sample structures of the ECO-G optimized sequences for $n = 31$, again showing the same trend. For odd values of $n$, there is presence of a loop with an odd number of residues or a single unpaired base at the end, but there is still a GC-rich double helix. In contrast, the structures generated by the under-performing algorithms do show presence of unpaired bases and are less in GC content, leading to high energy structures (e.g. Figures 8 and 9 are obtained via SA after 4000 steps for $n = 30$ and 60, respectively).

## E    DESIGN EXPERIMENTS

For our experiments, we focus on three puzzles from the Eterna-100 dataset Anderson-Lee et al. (2016). Two of the selected sequences (puzzles 15 and 41 of lengths 30 and 35, resp.), despite their fairly small lengths, are challenging for many algorithms (see Anderson-Lee et al. (2016)). In both puzzles, our MCTS variants (ECO-F and ECO-G) are able to significantly improve the performance of MCTS when limited number of true rewards are available. All algorithms outperformed RS as expected. Within the given 500 evaluation budget, both ECO-G, and especially ECO-F, are superior to LEARNA by a substantial margin (see Figure 12). In puzzle number 41 (Figure 13), again both ECO-G and ECO-F significantly outperform LEARNA, over the given number of evaluations. Interestingly, ECO-F is able to outperform LEARNA throughout the evaluation process, and in average finds a far better final solution than LEARNA.

The final sequence is puzzle #70 of length 184. The results of different algorithms over the latter puzzle is shown in Figure 14. As we can see from this figure, MCTS-RNA and LEARNA perform very similarly over the given 500 evaluation budget. ECO-F is able to outperform the remaining algorithm throughout the evaluation steps. Initially, ECO-G has a similar performance to those of

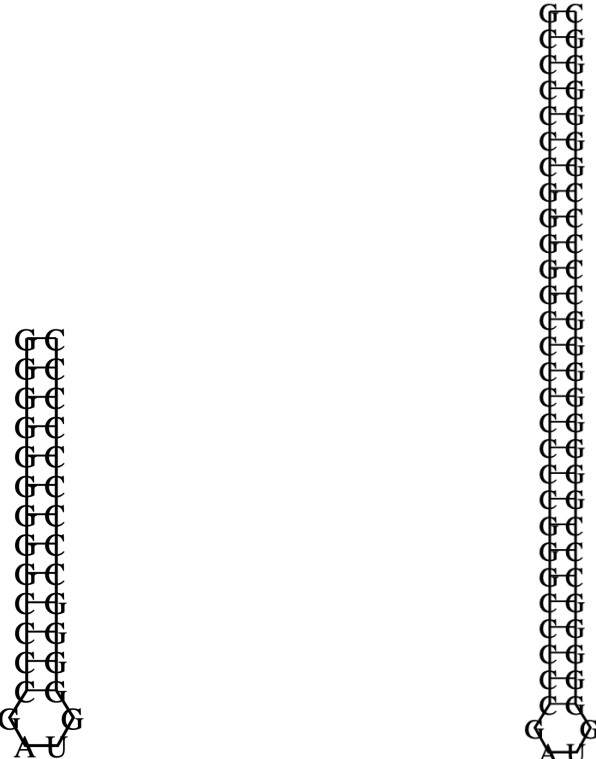

Figure 6: RNA Structure via ECO-G for $n = 30$    Figure 7: RNA Structure via ECO-G for $n = 60$

MCTS-RNA and LEARNA, but offers an improved performance over the latter two just after $400$ steps.

## F   CHOICE OF THE ACQUISITION METHOD

Throughout the experiments, we designated SA and MCTS as acquisition methods for generic BBO and design problems, respectively. The latter choice was made in accordance with the literature, where SA is typically used as a baseline and method of choice for the generic BBO problem, whereas MCTS has been commonly used for the design problem. For instance, SA has been considered as a baseline and/or acquisition method in Dadkhahi et al. (2020), Ricardo Baptista (2018), and Oh et al. (2019) (albeit with a different algorithm than ours). On the other hand, MCTS (i.e. RNA-MCTS as well as its variations) is perhaps the most popular RNA design technique in the literature. Here, we point out that both SA and MCTS can be used for both generic BBO and design problems. In this section, we compare the performance of different acquisition methods in each problem.

First, we consider the generic BBO problem of RNA sequence optimization with $n = 30$ (considered in Section 4). Figure 15 demonstrates the performance of ECO-F and ECO-G when SA or MCTS are used as acquisition methods. As we can see from this figure, the SA-as-acquisition-method variants perform slightly better than MCTS-as-acquisition-method counterparts over 500 steps. In particular, although the performance gap is initially moderately large, over time this performance gap becomes smaller.

Next, we consider two design problems considered in Section 4: puzzles #15 and #41. Note that, when using SA as the acquisition method, we apply the softmax operator (in Algorithm 1) over the set of $\{GC, CG, AU, UA\}$ if the corresponding variable is part of a paired base. As we can see from Figure 16, for puzzle #15, ECO-F with MCTS outperforms the remaining algorithms. The rest of the algorithms have very similar performances, with ECO-F (MCTS) marginally surpassing the SA variants. As we can see from Figure 17, for puzzle #41, the MCTS variant of ECO-F slightly

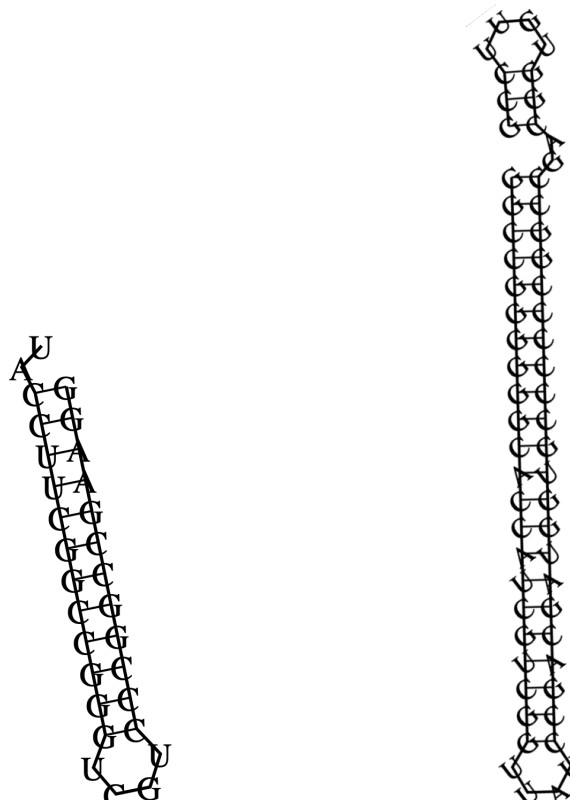

Figure 8: RNA Structure via SA for $n = 30$    Figure 9: RNA Structure via SA for $n = 60$

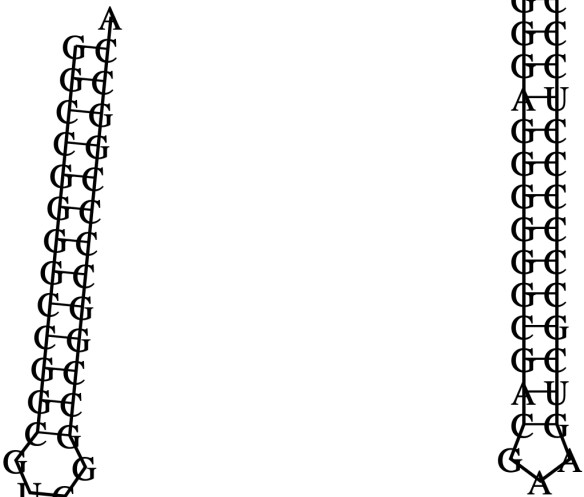

Figure 10: RNA Structure via ECO-G for $n = 31$    Figure 11: RNA Structure via ECO-G for $n = 31$

outperforms its SA variant, whereas the MCTS variant of ECO-G maintains a bigger gap with its SA variant.

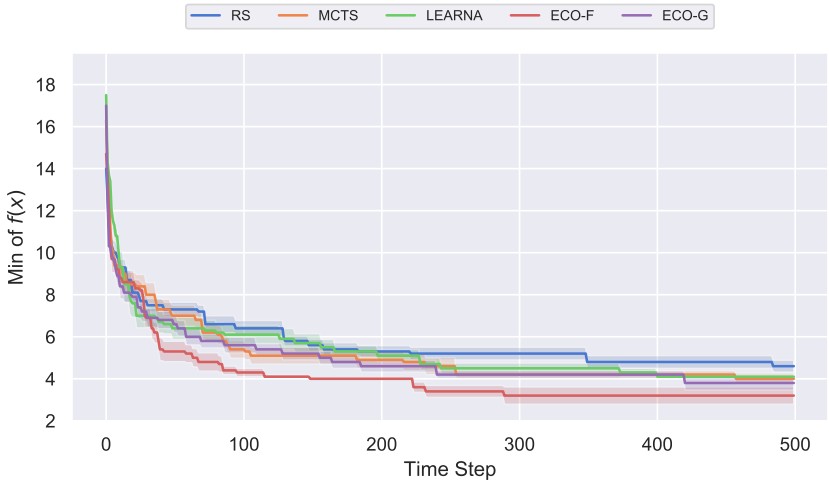

Figure 12: Best function evaluation for RNA Design of puzzle #15 with $n = 30$.

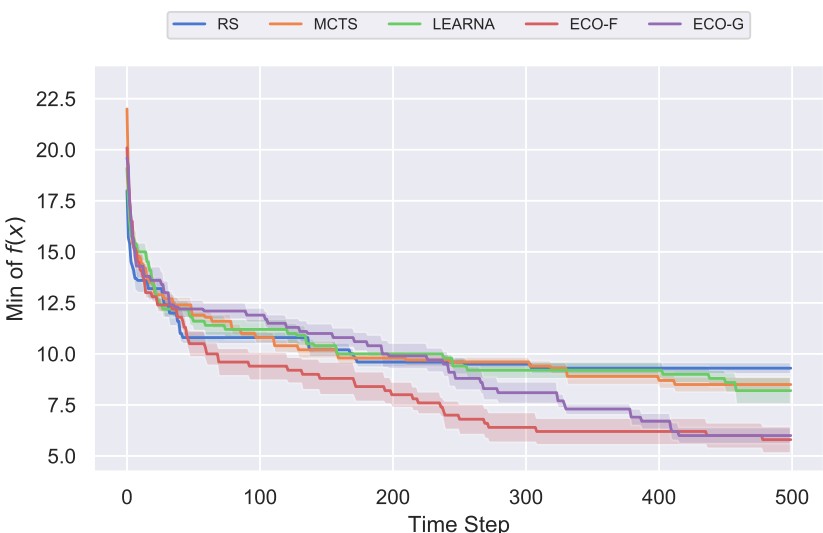

Figure 13: Best function evaluation for RNA Design of puzzle #41 with $n = 35$.

## G  ORDER OF THE SURROGATE MODEL

As mentioned in Section 4, we used $m = 2$ as the maximum order of the representations in all the experiments. In this section, we focus on the generic BBO problem of RNA optimization and investigate the impact of the model order on the performance of the proposed ECO algorithms. In particular, we compare the performance of the algorithm at $m = 3$ with that of $m = 2$. As we can see from Figure 18, at smaller evaluation budgets, the order 2 models moderately outperform the order 3 counterparts in both ECO-F and ECO-G. As we increase the number of samples, this performance gap becomes smaller. At the 500 evaluation budget, ECO-G3 outperforms ECO-G2 by a small margin of 0.1. At the same evaluation budget, ECO-F3 is slightly inferior to ECO-F2 by a margin of 0.2. Considering the convergence behavior of the curves at order 3 versus those of order 2, we expect the former models to eventually outperform the latter models at higher number of evaluations. However, since in BBO problems sample efficiency is typically of main concern, it would make sense to use low-order approximations. We point out that a similar observation was made in Ricardo Baptista (2018) for the Boolean case, where higher order models suffer from a slower start due to the higher

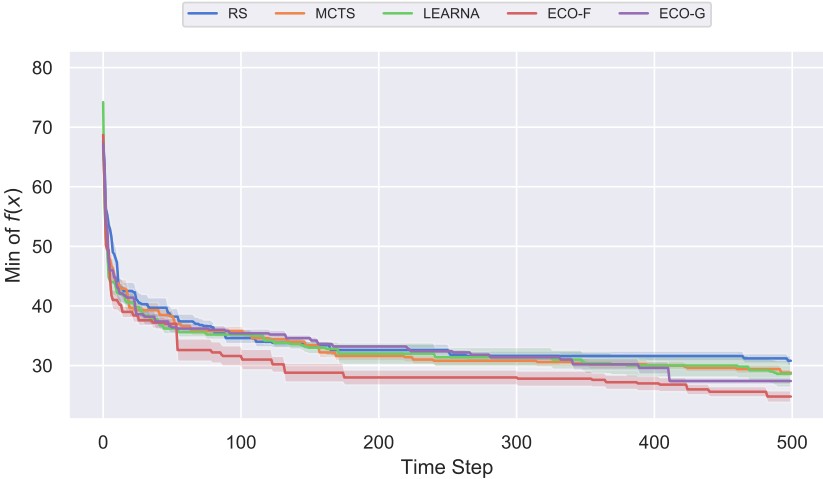

Figure 14: Best function evaluation for RNA Design of puzzle #70 with $n = 184$.

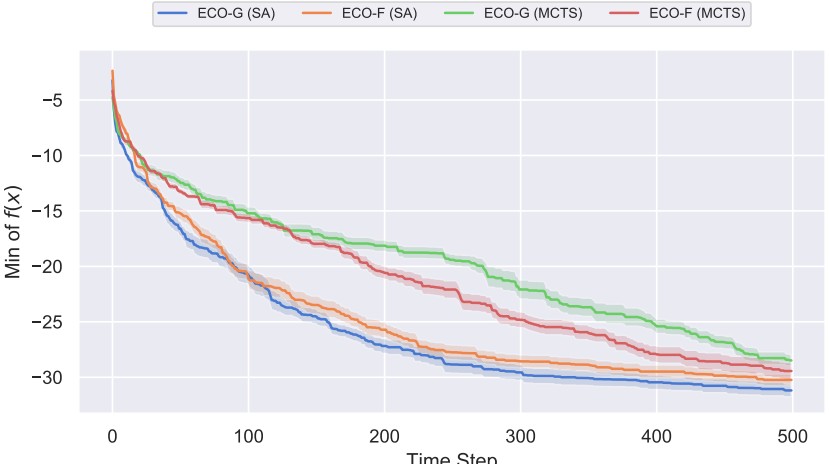

Figure 15: Comparison of different acquisition methods for the generic BBO problem of RNA sequence optimization with $n = 30$.

dimensionality of the parameter space. From our experiments in categorical problems, this behavior seems to be even more pronounced due to the higher dimensionality of the categorical domains.

A summary of the computation times for ECO algorithms at different model orders is given in table 2. Since the complexity of ECO is linear in the number of experts, which exponentially grows with the model order $m$, we observe an increase in the computational complexity of ECO-F3 (ECO-G3) versus that of ECO-F2 (ECO-G2) by a factor of 9.7 (16.3).

Table 2: Average computation time per step (in Seconds) at different model orders.

| ECO-F2 | ECO-F3 | ECO-G2 | ECO-G3 |
| --- | --- | --- | --- |
| 2.0 | 19.4 | 5.7 | 93.1 |

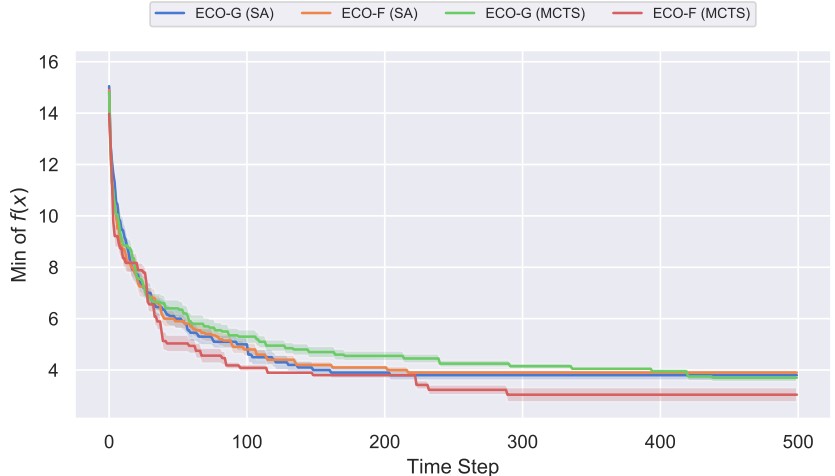

Figure 16: Comparison of different acquisition methods for design puzzle #15 with $n = 30$.

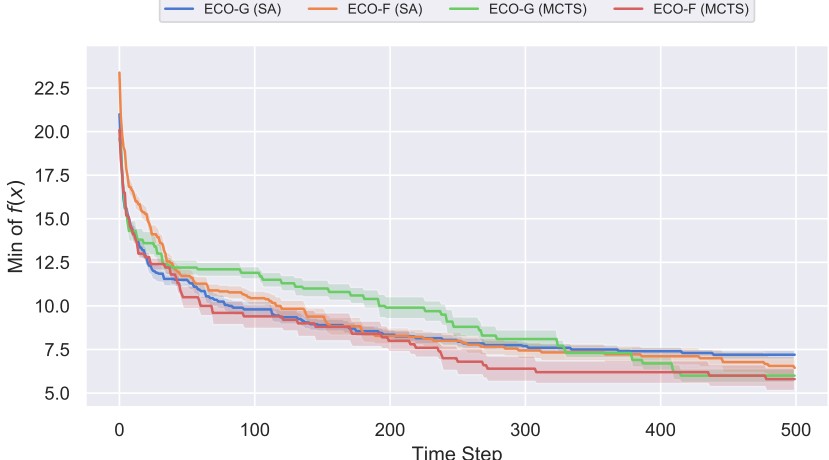

Figure 17: Comparison of different acquisition methods for design puzzle #41 with $n = 35$.

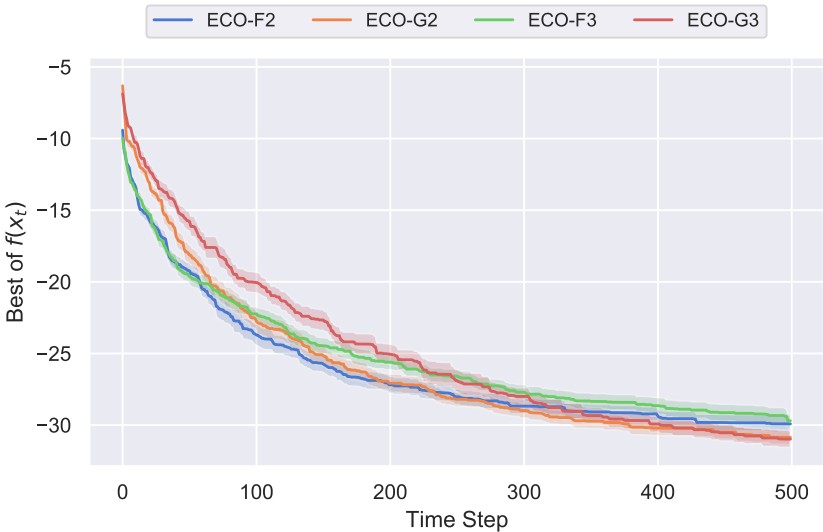

Figure 18: Impact of model order on the performance of ECO-F/G in the RNA optimization problem with $n = 30$.

