# OpenReview forum: "Fourier Representations for Black-Box Optimization over Categorical Variables"
_ICLR.cc/2021/Conference — Reject_

### Official Review · AnonReviewer3 · 2020-10-24
**The idea is interesting but there are some theoretical and empirical parts that need to be clarified more.**

**Rating:** 5
**Confidence:** 3

**Review:**

**Summary**
This paper proposes a model-based black-box function optimization on purely categorical variables. Two different representations for categorical variables are proposed, one is an improved pseudo-boolean function form capable of representing non-binary categorical variables in a compact way and another is to rely on (mathematical) group representation theory after mapping each categorical variable to a cyclic group. In acquisition function optimization, SA is used for generic BO and MCTS is used for design problems. The proposed EGD-F/G are compared with baselines and shown to have competitive computational efficiency with comparable performance.


**Strengths**
1. A compact representation for general (including non-binary) categorical variables is proposed and the strengths of the compact representation are theoretically supported, which is an improvement upon a bit ad-hoc approach to handle non-binary categorical variables proposed in BOCS.
2. The proposed algorithm is more efficient than COMBO with comparable performances. EGO-F/G may find more useful applications in problems where more evaluations are affordable than classical BO settings.


**Weaknesses**
1. EGO-F/G does not provide explicit uncertainty in its surrogate modeling, which is crucial in balancing exploitation-exploration trade-off. And the acquisition function is kind of just predictive mean of the surrogate model, it seems that balancing exploitation-exploration is the full responsibility of the acquisition function optimizer, which makes me think that the performance is attributed more to different choices of an acquisition function optimizer and less to surrogate models. However, if there is some stochasticity in surrogate model training, then this can be interpreted as Thompson sampling as NN trained with SGD can be regarded as a posterior sample, then it seems OK to say that surrogate model plays its part for exploitation-exploration trade-off. It would be good if the authors can clarify this.
2. Since, typically, not much information about an objective is available in advance, choosing a different acquisition function optimizer is not practical. Even though the test problems are divided into generic BBO and design problems, it seems that both SA and MCTS can be used in all experiments. For example, SA for the constrained problem can simply mask softmax values corresponding to invalid ones and MCTS for unconstrained ones is easier to adapt. If EGO-F/G is shown to be less sensitive to the choice of the acquisition function optimizer, then the argument for the strong representational power of the surrogate models will be supported more strongly. Therefore, it is recommended that the authors compare SA and MCTS in experiments where possible as BOCS(Ricardo Baptista, 2018) compares BOCS-SA and BOCS-SDP.
3. In spite of the benefits from Fourier transform on finite Abelian groups, giving a cyclic group structure to each categorical variable impose a random structure on values of the categorical variable. For example, if a categorical variable has 6 categories and is mapped to Z_6, then the categorical value corresponding to 2 and the categorical values corresponding to 4 are the inverse to each other due to the mapped group structure but this relation is not natural with regard to original categorical values. I was not able to find any theoretical/empirical analysis in the paper.
4. Up to the experiment section, the paper is presented in a way that the ultimate goal is to find an optimum as few evaluations as possible. However, in synthetic benchmarks, EGO-F/G are argued to be better than baselines because of the computational efficiency, which sounds a bit contradicting.


**Recommendation**
Because of the concern on the consistency of the performance from using different acquisition function optimizers, the empirical demonstration has points to be improved. And even though a mathematically elegant expert is given by Fourier transform on a finite abelian group, the correspondence between a categorical variable and a (finite) cyclic group needs more investigation. Therefore, in spite of its interesting idea, I think the paper needs some improvement for acceptance.


**Questions**
- SA and MCTS are acquisition function optimizers, not acquisition function itself. In contrast to typical BO where the acquisition function is a function of the predictive distribution given by the surrogate model, the acquisition function of EGO-F/G is the surrogate model itself (in other words, identity function on kind of predictive mean), isn't it?
- In regard to weakness 2, the correspondence between categorical values of a categorical variable and the elements of a group seems arbitrary. Does this mapping affect performance? Do you have any reasonable interpretation of this?
- In each experiment, which m (max model order) is used? Does the choice of m have a significant impact on runtime and optimization performance?
- Maybe on Eterna-100 dataset, COMBO is not applicable?


**Additional feedback** (Irrelevant to the decision assessment)
- The last sentence of the first paragraph of the introduction is a bit confusing. Since mixed variable problems include pure categorical ones as subproblems, intuitively, mixed variable problems look more challenging than purely combinatorial ones.
- It would be better if 'Proof, see appendix' is added under thm 3.1.

---

> ### Author Response · Authors · 2020-11-19
> **Response to Reviewer 3**
>
> > SA and MCTS are acquisition function optimizers ...
>
> That is correct. In the draft, we used a different definition for the acquisition function, by which we mean the module that selects a candidate point for black-box evaluation, given the current estimate for the surrogate model. As pointed out by the reviewer, our surrogate model is an estimate (at any given time t) rather than a predictive distribution. Please also see our reply to Reviewer 2 for a detailed formalization of various surrogate-acquisition frameworks.
>
> > Since, typically, not much information about an objective is available in advance, choosing a different acquisition function optimizer is not practical ...
>
> The rationale behind the designation of SA and MCTS as acquisition methods for generic BBO and design problems, respectively, is as follows. In the literature, SA is typically used as a baseline and method of choice for the generic BBO problem, whereas MCTS has been commonly used for the design problem. For instance, SA has been considered as a baseline and/or acquisition method in COMEX, BOCS, and COMBO (albeit with a different algorithm than ours). On the other hand, MCTS (i.e. RNA-MCTS as well as its variants) is perhaps the most popular RNA design technique in the literature.
>
> Having said that, we agree that both SA and MCTS can be used for both generic BBO and design problems. We added a new section on the choice of the acquisition methods in the appendix (Appendix F), comparing the performance of ECO-F/G when different acquisition methods (SA or MCTS) are used. In summary, for the RNA optimization problem (generic BBO), the SA variants slightly outperform the MCTS variants, although this performance gap seems to be decreasing over time. For the design problem, on the contrary, the MCTS variants surpass the SA variants. In the latter case, the results are more varied; in some cases, the performance gap is almost non-existent, whereas in others we observe a slight performance gap.
>
> Finally, we add that the representational power of the proposed surrogate models is demonstrated via the significant improvements obtained in the performance of ECO-F/G in conjunction with SA and MCTS versus those of vanilla SA and MCTS, respectively.
> Regardless of the representational power of the surrogate model, the search algorithm does impact the overall performance of any black-box optimization framework. The strong representational power of the surrogate model does not necessarily lead to identical performances when different search methods are used.
>
> > In regard to weakness 2, the correspondence between categorical values of a categorical variable and the elements of a group seems arbitrary ...
>
> Let $\chi = [k]^n$ be the categorical domain. Let the true function be $f$. For generality, let us consider a complex valued function $f: \chi \rightarrow \mathbb{C}$ where $\mathbb{C}$ is the field of complex numbers. The basis functions are $\psi_{{\cal I}} (x)$. Now, one can view a function as a $[k]^n$-length vector, one entry each for evaluating the function at every point in the domain $\chi$. We denote the vector for function $f$, thus obtained, by $f^{\chi} \in \mathbb{C}^{k^{n}}$. Similarly, denote the vector for evaluations of the basis function $\psi_{{\cal I}}$ by $\psi_{{\cal I}}^{\chi} \in \mathbb{C}^{k^{n}}$. Let $A$ be a matrix created by stacking all vectors corresponding to basis vectors in the columns. Then, the Fourier representation can be written as $f^{\chi} = A \alpha$ where $\alpha$ is the vector of Fourier coefficients in our group-theoretic representation. Now, due to the use of complex exponentials, one can show that $\sum_{x \in [k]^n} \psi_{{\cal I}} (x) \psi_{{\cal I}'} (x) = 0$ if ${\cal I} \neq {\cal I}'$. Therefore, the columns of the matrix $A$ are orthogonal. Hence, $A$ is a full rank matrix. Therefore, our representation is merely representing a vector in another full rank orthogonal basis. Hence, it is unique and complete. This is added to the appendix as a remark (Appendix A).
>
> In the example of group of integers modulo 6, although the numbers 2 and 4 are inverse of one another in the cyclic group defined over a given variable $x_i$ (assuming that $k = 6$), this does not impose any restriction on the values of the function at 2 and 6 for variable $x_i$. In other words, there is no connection between the values of $f$ at $x_i = 2$ and $x_i = 4$ given the values for the remaining variables $x_{-i}$. To be precise, the dimensionality of the representation (or the degree of freedom) is $k^n$ which exactly matches the function dimensionality.

---

> ### Author Response · Authors · 2020-11-19
> **Response to Reviewer 3 --- Continued**
>
> > In each experiment, which m (max model order) is used? ...
>
> We have used $m = 2$ in all the experiments (as mentioned in Section 4). We added a new section in the appendix (Appendix G) to compare the performance of order 2 and order 3 models in the RNA optimization problem. In summary, given the finite evaluation budget of $500$, we only observed minor changes in the performance of the algorithm when increasing the model order to 3. A similar observation was made in other problems.
>
> In the added section (Appendix G), we also included the computation times for both order 2 and order 3 models. As discussed in Section 3, the computational complexity of ECO is linear in the number of experts, which in turn grows exponentially with order $m$.
>
> We further add that we have already developed pruning strategies to select a subset of experts automatically (particularly relevant for problems over higher order models and/or larger numbers of variables), which we did not include in this paper due to space limitations. We added this as an additional supplement (can be added to the appendix if necessary).
>
> > Up to the experiment section, the paper is presented in a way that the ultimate goal is to find an optimum as few evaluations as possible ...
>
> In all the experiments, the performance of ECO-F/G have been compared against the baselines as well as state-of-the-art methods in terms of sample efficiency. All the plots shown in the manuscript/appendix depict the minimum function value (y-axis) found versus the number of black-box evaluations (x-axis). The computational advantage of ECO-F/G with respect to COMBO (and Bayesian optimization methods in general) comes as an added bonus, which makes it computationally tractable for problems over larger numbers of variables.
>
> > Maybe on Eterna-100 dataset, COMBO is not applicable?
>
> We compared the performance of our algorithms in design problems against a state-of-the-art algorithm specifically developed for that problem, i.e. LEARNA for the RNA design problem. While COMBO, in theory, can be used for the design problem, after some modifications, we are not aware of any such effort in the literature. The major obstacle in utilizing Bayesian optimization methods in general, and COMBO in particular, for design problems is their high computational cost, which makes them impractical for problems over even moderate sequence lengths (let alone larger sequence lengths), which are typically of interest in design problems.

---

### Official Review · AnonReviewer4 · 2020-10-28

**Rating:** 6
**Confidence:** 3

**Review:**

Paper Summary

The paper considers the problem of black-box optimization of expensive functions defined over categorical variables. A surrogate model-based optimization approach is proposed to tackle this problem. Fourier representations are proposed as surrogate model by treating the categorical input as the direct sum of cyclic groups Z/kZ (k is the arity/category size). The coefficients of this representation are learned via exponentially-weighted update rule. The selection of each subsequent input for evaluation is performed via direct optimization of the surrogate model built over inputs collected previously. Simulated Annealing and Monte Carlo Tree Search is proposed as the acquisition function optimization procedure for unconstrained and constrained problems respectively. Experiments are performed on two synthetic problems and RNA-sequence optimization.

Detailed Comments

- The paper considers an important problem which has multiple applications (for e.g. biological sequence design) in practice.

- Although the proposed method is a natural generalization of the COMEX [5] approach which was proposed for binary variables, it shows good performance on two benchmarks and can be useful for end users because of its simplicity.

- Regarding the two proposed representations, isn't the first one (Equation (2)) a special case of the second (Equation (6))? What are the tradeoffs in choosing one among them?

- There are many important and relevant related work (both for surrogate modeling and acquisition function optimization) that are not discussed in the paper.  Please provide a detailed discussion comparing proposed approach with the below methods otherwise it comes across as if there is limited existing work for the considered problem. They are very relevant to the paper because all of them consider the setting of "small" data with expensive function evaluations.
  - Surrogate modeling
	- SMAC [1] is the most natural approach that handles categorical variables nicely.
	- Tree structured Parzen Estimator (TPE) [2] is another approach that can easily handle categorical variables.
	- Walsh functions [3] have been used effectively for surrogate modeling over discrete variables.
  - Acquisition function optimization
  	- Amortized Bayesian Optimization over Discrete Spaces [4]

- The proposed one-hot encoding in Equation (2) is said to have "far less terms than a vanilla encoding". Please provide a quantitative description of this reduction of number of terms.

- It is not entirely clear how the representation for a real-valued function reduces to (8) from (6). Do we just ignore the complex part similar to the common approach in random Fourier features used for kernel methods? Please provide a clear derivation.

- Is the choice of n=30 for RNA sequence optimization motivated by some real-world implication?

References

[1] Hutter, F. and Hoos, H. H. and Leyton-Brown, K. Sequential Model-Based Optimization for General Algorithm Configuration In: Proceedings of the conference on Learning and Intelligent OptimizatioN (LION 5)

[2] Bergstra, J. S., Bardenet, R., Bengio, Y., & Kégl, B. (2011). Algorithms for hyper-parameter optimization. In Advances in neural information processing systems (pp. 2546-2554).

[3] Leprêtre, F., Verel, S., Fonlupt, C., & Marion, V. (2019, July). Walsh functions as surrogate model for pseudo-boolean optimization problems. In Proceedings of the Genetic and Evolutionary Computation Conference (pp. 303-311).

[4] Swersky, K., Rubanova, Y., Dohan, D., & Murphy, K. (2020, August). Amortized Bayesian Optimization over Discrete Spaces. In Conference on Uncertainty in Artificial Intelligence (pp. 769-778). PMLR.

[5] Dadkhahi, H., Shanmugam, K., Rios, J., Das, P., Hoffman, S., Loeffler, T. D., & Sankaranarayanan, S. (2020). Combinatorial Black-Box Optimization with Expert Advice. arXiv preprint arXiv:2006.03963.

---

> ### Author Response · Authors · 2020-11-19
> **Response to Reviewer 4**
>
> > Regarding the two proposed representations, isn't the first one (Equation (2)) a special case of the second (Equation (6))? What are the trade-offs in choosing one among them?
>
> The two representations are identical only at $k = 2$ (i.e. the Boolean case). In general, since the terms (experts) in the first representation are monomials, whereas the terms in the second representation are sines and cosines (or complex exponentials), the two representations are completely different.
>
> As we can see from experiments, in some experiments the first representation outperforms the second one, whereas in others we have the opposite scenario. The answer to the question of which representation is better for a given problem is not trivial, and depends on the properties of the black-box function at hand. As a result of this, one interesting direction for future research would be to devise an ensemble model, which would maintain both representations, and would ideally perform at least as well as the best one (or even better than both); this is also pointed out in the Future Work Section.
>
> > There are many important and relevant related work (both for surrogate modeling and acquisition function optimization) that are not discussed in the paper ...
>
> We thank the reviewer for pointing out the missing citations. We added the following descriptions to the Related Work section in the draft.
>
> - [1] suggests a surrogate model based on random forests to address optimization problems over categorical variables. The proposed SMAC algorithm uses a randomized local search under the expected improvement acquisition criterion in order to obtain candidate points for black-box evaluations. [2] suggests a tree-structured Parzen estimator (TPE) for approximating the surrogate model, and maximizes the expected improvement criterion to find candidate points for evaluation.
> For optimization problems over Boolean variables, multilinear polynomials [BOCS, COMEX] and Walsh functions [3] have been used in the literature.
>
> - As an alternative to parameter free search methods (such as SA), [4] suggests to use a parameterized policy to generate candidates that maximize the acquisition function in Bayesian optimization over discrete search spaces. Our MCTS acquisition method is similar in concept to [4] in the sense that the tabular value functions are constructed and maintained over different time steps. However, we are maintaining value functions rather than a policy network.
>
> > The proposed one-hot encoding in Equation (2) is said to have "far less terms than a vanilla encoding". Please provide a quantitative description of this reduction of number of terms.
>
> When all the terms up to max degree of $n$ are used, the number of terms in vanilla one-hot representation is $2^{kn}$, whereas our representation reduces this number to $k^n$ matching the space dimensionality, thereby making the algorithm computationally tractable and efficient. As an example, in the RNA optimization problem with $k = 4$ and $n = 30$, this leads to a reduction of terms by a whopping factor of $\approx 1.267 \times 10^{30}$. As stated in Theorem 1, despite this significant reduction, the resulting representation is in fact unique and complete, and we are not losing any information in the process.
> When a max degree of $m$ is used in the approximate representation, the number of terms (as mentioned in the appendix) in our proposed representation is equal to $d = \sum_{i=0}^m \binom{n}{i} (k-1)^i$. This number in a vanilla one-hot encoded representation would be equal to $\sum_{i=0}^m \binom{nk}{i}$. We added the full description on the number of terms to the appendix (Appendix B).
>
> > It is not entirely clear how the representation for a real-valued function reduces to (8) from (6) ...
>
> Since the black-box function is assumed to be real-valued, the imaginary part of the function has to be zero. As a result, we discard the imaginary part of the representation, and only consider the real part.
>
> > Is the choice of n=30 for RNA sequence optimization motivated by some real-world implication?
>
> From a biological standpoint, this is the max length of short RNA sequences that play a number of regulatory roles from plants to animals, which imply their involvement in fundamental cellular processes (see D. P. Bartel, “MicroRNAs: genomics, biogenesis, mechanism, and function,” Cell, vol. 116, no. 2, pp. 281–297, 2004.).
>
> In our experiments, we picked the sequence length of $n=30$ since COMBO could be run in about $24$ hours (for comparison purposes). We note that for higher sequence lengths, COMBO becomes computationally impractical, and this is one of the advantages of our proposed ECO algorithm.

---

### Official Review · AnonReviewer2 · 2020-10-28
**The authors offer a Fourier representation for categorical variables that allows for generalization of observations of the response surface across categories.**

**Rating:** 6
**Confidence:** 1

**Review:**

This seems like a very interesting paper and perhaps quite impactful indeed, if it achieves what it claims to. Unfortunately, assessing the novelty and merit of this paper relies heavily on expertise in Group theory, which I certainly lack. Here are some scattered thoughts, for whatever they're worth.

On the one hand, assuming a surrogate function that is equipped with this ability to generalize, leveraging MCTS and the UCT selection criterion as an acquisition function seems reasonable to me. On the other hand, it seems to me that using SA, targeting a tempered surrogate, might be too greedy and not align with the latest approaches in black box optimization, where some measure of uncertainty is used in the acquisition decision-making process. It would be good to have a discussion on how one could obtain an uncertainty quantification from the decomposition (e.g. uncertainties around each coefficient alpha and how that extends to uncertainty in f).

In terms of experimentation, since this paper introduces a new decomposition that is complete and unique, I would've expected to see some results concerning how well a truncated decomposition fits a known function of categorical variables. Some toy experiments would provide valuable evidence that the surrogate learned from such a truncated decomposition is likely good even for relatively short truncations.

As I cannot pass judgement on the novel aspects of this paper, I will be generous with my score and let my confidence score reflect my lack of expertise. Looking forward to reading other reviews and comments on this manuscript.

---

> ### Author Response · Authors · 2020-11-19
> **Response to Reviewer 2**
>
> > On the one hand, assuming a surrogate function that is equipped with this ability to generalize ...
>
> (We added a summary of the following insights to the Future Work Section, i.e. Section 5.)
>
> To answer this question let us consider three acquisition strategies a) UCB b) Thompson Sampling and c) Our SA Algorithm.
>
> Let $p_t(w)$ be the posterior distribution on the linear coefficients in $\textit{any}$ of our representations with respect to a prior $p_0(w)$. Suppose $p_t(w)$ is obtained using a standard Bayesian inference procedure given past data and the corresponding evaluations. In fact, for the Boolean case, the BOCS algorithm operates in this setting. Let $m_t[x]$ be the mean function due to the posterior (which is a function of the categorical vector $x$). Let $\sigma_t[x]$ be the standard deviation under the posterior $p_t$ at a point $x$. Let $f_w[x]$ be the categorical function when weight $w$ is used in the representation. Let $f^h_t(x)$ be our hedge surrogate at time $t$.
>
> The three acquisition strategies will look as follows:
>
> $\textbf{UCB}$: $\mathrm{argmax}_{x \in [k]^n } m_t[x] + \gamma_t \sigma_t[x]$.
>
> $\textbf{TS}$: Sample $w \sim p_t(\cdot)$. Then compute $\mathrm{argmax}_{x \in [k]^n} f_w(x)$.
>
> $\textbf{Our SA}$: $\mathrm{argmax}_{x \in [k]^n} f^h_t(x) + \gamma_t n(x)$ where $n(x)$ is sampled i.i.d for every $x$ from a Gumbel Distribution.
>
> We are using the property that Gibbs sampling over discrete domain is equivalent to logistic sampling which can be done using Gumbel softmax.
>
> In $\textit{all}$ the above, the acquisition strategy involves an optimization over the categorical domain of some categorical function which itself is a hard problem. Hence, it is non-trivial to find the argmax in the UCB and TS cases as each leads to another combinatorial optimization problem over a categorical domain. Please note that the argmax in our SA method is approximated via Gibbs sampling. In fact, SA could be used further to approximate it by adding a further uncertainty term! Representation of $f_w[x], m_t[x], \sigma_t[x]$ is a crucial component even if one were to contemplate combinatorial optimization routines for categorical domains.
>
> In our case, we don't use a Bayesian posterior mean function but rather an online approximator learnt using Hedge. Hedge algorithm has $\textit{strong adversarial guarantees}$ (please see the COMEX paper for theoretical results in the Boolean case). In other words, given any additional black box evaluation, it is guaranteed to move closer to the true black-box model in some distance; This carries over to our setting as well. However, there is a $\textit{domain independent}$ exploration bonus due to $n(x)$ being sampled i.i.d from the same distribution regardless of $x$. The terms that account for uncertainty in both TS and UCB depend on $x$.
>
> In conclusion, domain dependent uncertainty incorporation is left for future work. However, it is non-trivial to do so efficiently due to the argmax operation over a categorical domain. Our main contribution is the representations of the surrogate model which is learnt using hedge in an adversarial setting. We believe our contribution would form the basis for further work irrespective of the framework adopted (as can be seen above).
>
> Finally, we point out that the applicability of the proposed representations is not limited to the surrogate model learning with expert advice (i.e. ECO). For instance, the proposed representations can be used in a BOCS-type algorithm with other acquisition frameworks like UCB and TS.
>
> > In terms of experimentation, since this paper introduces a new decomposition that is complete and unique, I would've expected to see some results concerning how well a truncated decomposition fits a known function of categorical variables ...
>
> As suggested by the reviewer, we added a section in the appendix (Appendix G), comparing the performance of order 2 and order 3 models for both ECO-F and ECO-G in the RNA optimization problem. Please also see our responses to Reviewers 1 and 3.

---

### Official Review · AnonReviewer1 · 2020-10-29

**Rating:** 6
**Confidence:** 3

**Review:**

The paper proposes two representations, namely one-hot encoded Boolean expansion and group-theoretical Fourier expansion, for the surrogate model used for the black-box evaluations on purely categorical variables. With the two surrogate models, the authors tackle both the black-box optimization problem and the design problem. Two forms of acquisition functions are applied for query selection – simulated annealing and Monte Carlo Tree Search. The new algorithms are compared with the existing methods in simulations and have advantage for the objective value and speed-up.

The paper has many contributions, and I would be inclined to recommend the paper for acceptance, after the rebuttal.

My concerns are three-folds.
1. The paper seems not to have clearly abundant novelty, as far as I understand. The author should highlight what is new for one-hot encoded function and group-theoretical Fourier expansion. Also, the simulated annealing and Monte Carlo Tree Search are not fully novel and I seem not to fully see the major changes to these methods. It would be great if the authors could list the major changes to these algorithms to fully show novelty.
2. The one-hot Boolean expansion could have a certain degree of lack of scalability if the order of approximation is large. If the order $m$ has a large value, then all $m$-subset of set $[n]$ might have too many terms for the representation to be efficient. A method to prune the $m$-subset should be needed.
3. The one-hot encoded Boolean function and group-theoretical expansion should have applications that they fit well and problems that they are less applicable. It seems critical to identify the superb properties of these functions than other alternatives, and a rough range of the problems that work better with these surrogate functions. Without such identification, the surrogate functions have reduced significance.

Please fix the following unimportant typos. The variable $j$ is overloaded in Eq (7), as it is both the complex number unit and the integer pair.  The r and i subscript in Eq (8) are not defined, for real and imaginary part of the function $f_{\alpha}(x)$.

---

> ### Author Response · Authors · 2020-11-19
> **Response to Reviewer 1**
>
> > The paper proposes two representations, namely one-hot encoded Boolean expansion ...
>
> We propose an abridged version of one-hot encoded Boolean Fourier expansion rather than the vanilla one-hot encoded Boolean Fourier expansion, and this is novel to our work, to the best of our knowledge.
>
> > The paper seems not to have clearly abundant novelty  ...
>
> The novelty of our work (in part; see Contributions in Section 1 for a complete list) is to propose two representations for modeling functions over purely categorical variables. Such functions are of wide interest in the context of real-world applications, e.g. in the design and optimization of chemical or biological molecules.
> Of the two representations we propose, the abridged one-hot encoded Fourier representation is novel to this work; Fourier representation on Abelian groups has not been previously used as a surrogate model for functions over purely categorical variables. The number of terms in vanilla one-hot representation (typically used in literature to convert categorical problems to Boolean ones) is $2^{kn}$, whereas our representation reduces this number to $k^n$ matching the space dimensionality, thereby making the algorithm computationally manageable and efficient. This leads to a unique and complete representation -- superior/competitive performance of ECO is shown in experiments with up to $100\times$ speed-up with respect to COMBO.
>
> We believe that the applicability/significance of the proposed representations go far beyond the framework of black-box optimization with expert advice and can be used in other optimization algorithms and contexts.
> One avenue is to develop a BOCS-like algorithm for optimization over categorical variables using our representations.
> Another avenue is applications in reinforcement learning (RL): given that the elements of action and state spaces can be expressed as vectors of categorical variables, our representations can be used $(i)$ in order to model reward functions in model-based RL and $(ii)$ as linear value function approximators.
>
> > The one-hot Boolean expansion could have a certain degree of lack of scalability if the order of approximation is large ...
>
> We agree with the reviewer that utilizing one-hot encoded representations in order to convert Boolean representations to categorical ones is not scalable and would lead to too many terms. In fact, this is exactly the motivation behind our proposed representations. The main idea of our first representation is to overcome such shortcomings in one-hot encoding. We introduce an abridged version of one-hot encoded Boolean Fourier expansion, where the number of terms has been significantly reduced in comparison to the vanilla counterpart (please see our response to Reviewer 4 and a detailed description of the number of experts in Appendix B). Despite this significant reduction, we prove that the resulting representation is in fact unique and complete. Our second representation provides an alternative where we define a group structure among categorical variables to avoid one-hot encoding completely. To the best of our knowledge, both representations for modeling black-box functions over categorical variables are novel to this work.
>
> We further add that we have already developed pruning strategies to select a subset of experts automatically (particularly relevant in problems over higher order models and/or larger numbers of variables), which we did not include in this paper due to space limitations. We added this as an additional supplement (can be added to the appendix if necessary).
>
> > The one-hot encoded Boolean function and group-theoretical expansion should have applications that they fit well and problems that they are less applicable ...
>
> In general, both proposed representations are exact and complete. As such, both representations would fit any function over purely categorical variables. In our experiments, we have truncated the representations to a max degree of two. In many applications, a low-order model is sufficient to capture the interactions among different variables (and are naturally a better fit to our representations). Although this could potentially lead to approximation errors in some applications where higher-order interactions are present, this typically allows for trading off approximation accuracy for scalability.
> We add that, in existing physics-based functions the majority of energy is concentrated in lower-order terms, rendering low-order approximations justifiable. As a result, in practice, it is highly unlikely/rare to see any benefits at $m \geq 4$. Finally, we point out that similar observations were made in BOCS and COMEX papers for the Boolean problem.
>
> We added a Section in the appendix (Appendix G), where we compare the performance of order 2 and order 3 models for the RNA optimization problem. Please also see our response to Reviewer 3.

---

### Author Response · Authors · 2020-11-25
**Response Overview**

We would like to thank all the reviewers for evaluating our submission. We responded in detail to all the comments. In summary, we made the following changes to the manuscript:

1. We added experiments on the impact of the surrogate model order (Appendix G).
2. We added experiments on the choice of the acquisition method, i.e. MCTS versus SA (Appendix F).
3. We added a detailed quantification of the number of terms (experts) in each representation (Appendix B).
4. We provided proof that the proposed group-theoretic Fourier representation is unique and complete (Appendix A).
5. We expanded our related work section to include important additional references on surrogate models and acquisition methods.
6. We provided insights on uncertainty quantification in our algorithm as compared to TS and UCB. A summary of this is added to the Future Work Section and is left for further research.
7. We offer an algorithm for pruning experts, which is particularly relevant in problems over higher order models and/or larger numbers of variables (Provided as an additional supplement; can be added to the appendix if necessary).

We would be happy to make further additions/corrections if necessary. In conclusion, we believe that our paper and our proposed representations are not only of interest in both combinatorial black-box optimization and biological sequence optimization/design, but also would find applications in other problems involving functions over categorical variables.

---

### Decision · Program_Chairs · 2021-01-07
**Final Decision**

**Decision:**

Reject

**Comment:**

This paper considers the problem of black-box optimization over categorical variables using expensive function evaluations.
- Fourier representation is proposed as surrogate model by treating the categorical input as the direct sum of cyclic groups. The parameters are learned using exponentially-weighted update algorithm.
- To select the inputs for evaluation, simulated annealing and MCTS are employed as search algorithms to optimize the learned surrogate function.
- Experiments are performed on two synthetic problems and RNA sequence design problems.

The proposed fourier representation is novel and the results show the promise of this method in terms of computational-efficiency over state-of-the-art COMBO method.

There are two unsatisfactory aspects of this paper.
1. In expensive black-box optimization problems, number of function evaluations to find better solutions is critical. This paper takes a non-Bayesian approach to improve computational-efficiency (over prior Bayesian optimization methods), but this same advantage comes at the expense of sample-efficiency (number of function evaluations) due to lack of exploration.
2. In fourier representation, mapping categorical values to different group elements may change which basis are used for modelling. From a practitioner's perspective, it is important to verify that the performance is not significantly affected by this choice. This can be verified empirically. Even though one reviewer raised this point, authors' haven't responded though it is an easy experiment to do.

Due to the above shortcomings, the paper is judged to be not ready for publication at the current stage. I strongly encourage to resubmit the paper after addressing the above two concerns.